

**Develop a coupled agent-based modeling approach for**
**uncertain water management decisions**
Jin-Young Hyun[1], Shih-Yu Huang[1], Y. C. Ethan Yang[1*], Vincent Tidwell[2] and Jordan
Macknick[3]
[1]Lehigh University, Bethlehem, Pennsylvania
[2]Sandia National Laboratories, Albuquerque, New Mexico
[3]National Renewable Energy Laboratory, Golden, Colorado
([*]Corresponding Author: yey217@lehigh.edu; +1-610-758-5685)
## Abstract
Managing water resources in a complex adaptive natural-human system is subject to a
challenging task due to the difficulty of modeling human behavior and decision uncertainty. The
interaction between human-engineered systems and natural processes needs to be modeled
explicitly, and a formal approach is required to characterize human decision-making processes and
quantify the associated uncertainty caused by incomplete/ambiguous information. In this study,
we "two-way" coupled an agent-based model (ABM) with a river-routing and reservoir
management model (RiverWare) while ABM uses a bottom-up approach that allows individual
decision makers to be defined as agents – each able to make their own decisions based on their
objectives and confidence in the acquired information. The human decision-making processes is
described in the ABM using Bayesian Inference (BI) mapping joined with a Cost-Loss (CL) model
(BC-ABM). Incorporating BI mapping into an ABM allows an agent's internal (psychological)
thinking process to be specified by a cognitive map between decisions and relevant preceding
factors that could affect decision-making. The associated decision uncertainty is characterized by
a risk perception parameter in the BI mapping representing an agent's belief on the preceding
factors. Integration of the CL model addresses an agent's behavior caused by changing
socioeconomic conditions. We use the San Juan River Basin in New Mexico, USA to demonstrate
the utility of this method. The calibrated BC-ABM-RiverWare model is shown to capture the
dynamics of historical irrigated area and streamflow changes. The results suggest that the proposed
BC-ABM framework provides an improved representation of human decision-making processes
compared to conventional rule-based ABMs that does not take uncertainties into account. Future
studies will focus on modifying the BI mapping to consider direct agents' interactions, up-front
cost, joint human and natural uncertainty evaluation, and upscaling the watershed ABM to the
regional scale.
**Keywords**: Risk perception, Bayesian Inference Mapping, Cost-Loss Model, Coupled natural-
human systems, Energy-Water Nexus



## 1. Introduction

36  Managing water resources for growing demands of energy and food while sustaining the

37 environment is a grand challenge of our time, especially when we are dealing with a complex

38 adaptive natural-human system that subject to various sources of uncertainties. Nowadays, almost

39 every major basin in the world can be considered as a coupled natural-human system (CNHS)

40 where heterogeneous human activities are affecting the natural hydrologic cycle and vice versa

41 (Liu et al., 2007). The interaction between human activity and the natural environment needs to be

42 explicitly addressed, and the uncertainty within this complex system characterized according to a

43 formal approach if benefits toward improved water resource management (Brown et al., 2015) are

44 to be realized.

45  Recently, agent-based modeling (ABM) has become a commonly used tool in the scientific

46 community to address CNHS issues. An ABM framework identifies individual actors as unique

47 and autonomous "agents" that operate according to a distinct purpose. Agents follow certain

48 behavioral rules and interact with each other in a shared environment. By explicitly representing

49 the interaction between human agents (e.g., farmers) and the environment (e.g., a watershed) where

50 they are located, ABM provides a natural "bottom-up" setting to study transdisciplinary issues in

51 CNHS. Applying ABM approach in water resources management began a decade ago and became

52 a popular topic in CNHS analyses (Berglund, 2015; Giuliani et al., 2015; Giuliani and Castelletti,

53 2013; Hu et al., 2017; Khan et al., 2017; Mulligan et al., 2014; Schlüter et al., 2009; Yang et al.,

54 2009; Yang et al., 2012; Zechman, 2011).

55  However, one major challenge of applying ABM approach to water management decisions

56 arise from the difficulty of adequtely characterizing human decision-making processes and meet

57 real-world management intuition. The traditional approach through, for example, survey or



interview with local decision makers is extremely limited (e.g., Manson and Evans, 2007) in space
and time. Therefore, this study introduces the Theory of Planned Behavior (TPB), a well-known
theory in psychology used to predict human behavioral intention and actual behavior (Ajzen, 1991),
into ABM framework to quantify human decision-making processes. The TPB states that an
individual's beliefs and behaviors can be expressed in terms of a combination of attitude toward
behavior, subjective norms, and perceived behavioral control. Attitude toward behavior and
subjective norms specify an individual's perceptions of performing a behavior affected by its
internal thinking processes and social normative pressures, while perceived behavioral control
describes the effects from external uncontrollable factors (e.g., socioeconomic conditions). If an
individual has high belief about making a specific decision, then it has an increased confidence
that s/he can perform the specific behavior successfully. On the other hand, the tendency of a
person for making a specific decision increases/decreases if social normative pressures
decrease/increase.

Implementating the TPB into ABM requires that all the three components to be modeled

explicitly. In this study, we adapt the Bayesian Inference (BI) mapping (Pope and Gimblett, 2015)
and the Cost-Loss model (CL) (Thompson, 1952) for this task. The BI mapping (also called
Bayesian networks, belief networks, Bayesian belief networks, causal probabilistic networks, or
causal networks), built on the Bayesian probability theory and cognitive mapping, calculates the
likelihood that a specific decision will be made (Sedki and de Beaufort, 2012 via Pope and
Gimblett, 2015) while sequentially updating beliefs of specific preceding factors (model
parameters) as new information is acquired (Dorazio and Johnson, 2003). By applying the BI
mapping, an individual's beliefs affected by its internal thinking processes and perceptions of
social normative pressures can be described as a cognitive map between decisions and relevant



preceding factors. Ng et al. (2011) developed an ABM using BI to model the farmer's adaptation
of their expectations (or belief) and uncertianties of future crop yield, cost, and weather. Yet the
preceding factors were assumed to be independent of each other, which is not always true
especially if two preceding factors are spatially related (e.g., downstream reservoir elevation and
upstream precipitation). More importantly, the internal thinking processes of all farmers were
assumed to be the same (i.e., no spatial heterogeneity is modeled). As a result, a more realistic
framewok of applying BI to ABM is still needed to improve representation of human decision-
making processes.

While BI mapping specifies the human psychological decision-making process, CL model

addresses the effect of external socioeconomic conditions on an individual's decision-making (i.e.,
perceived behavioral control in the TPB). CL model is frequently used as a simple decision-making
model in economic analysis to quantify human decision-making according to economic theory
(Thompson, 1952). CL modeling has been widely used in estimating the economic value of
weather forecasts (Keeney, 1982; Lee and Lee, 2007; Murphy, 1976; Murphy et al., 1985). Tena
and Gómez (2008) and Matte et al. (2017) incorporated the Constant Absolute Risk Aversion
theory in CL modeling to evaluate risk perception of decision makers since the original CL model
assumes a risk-neutral decision maker. They used a parameter, Arrow-Pratt coefficient, to
represent "risk-averse" and "risk-seeking" decision makers but did not specify how this parameter
could be determined. They also did not clarify what will happen if different decision makers in the
system have different perceptions of risk (again, no spatial heterogeneity).

Evaluating uncertainty in CNHS is another challenge. For example, uncertainties involve

in the water resources decision making include errors in measurement and sampling of natural
systems, environmental variability, or incomplete knowledge of others behavior (Dorazio and



Johnson, 2003). Previous studies have demonstrated that quantitative information of uncertainty
can facilitate water resouce management in terms of selecting strategies, reduce implementation
cost, and adapt more effectively to unexpected changes in circumstances (e.g., Singh et al., 2010a).
Uncertainty in water resource management can be divided into two basic terms: variability and
ambiguity (Vucetic and Simonović, 2011). The variability describes the uncertainty in relation to
the inherent physical characteristics of water resources systems (i.e., hydrologic variability), while
ambiguity is the uncertainty in human decision-making processes caused by a fundamental lack of
knowledge or ambiguous information (Simonović, 2009).

Efforts of quantifying uncertainty in water resources management intensified in the 1980s

(Rogers and Fiering, 1986). Given the difficulty of modeling human behavior and decision
uncertainty (Loucks, 1992; Schlüter et al., 2012), previous studies have largely focused on
characterizing uncertainties associated with hydrologic variability such as climate (Hall et al.,
2012), surface water (Herman et al., 2014) and groundwater (Singh et al., 2010b). Optimal
management schemes like robust decision making (Lempert and Collins., 2007) and decision
scaling (Brown et al., 2012) have been developed to address uncertainties common to the natural
environment. In contrast, only a handful of existing studies adequately addresses the human
decision uncertianty caused by incomplete or ambiguous information. Quantifying these
uncertainties faces the fundamental challenge of understanding how the brain combines "noisy"
sensory information with prior knowledge to perceive an act in the natural world (Huang et al.,
2012). As a result, the human decision uncertainty caused by ambiguous or incomplete knowledge
has been either neglected or simplified and remain a vital issue for sustainable water resources
management (Fulton et al., 2011; Schlüter et al., 2017).





126   To address all these research gaps aforementioned, we developed an ABM based on the BI

127 mapping and CL model, as an implementaiton of the TPB, and referred to as the BC-ABM. The

128 BC-ABM is "two-way" coupled with a river-routing and reservoir management model (RiverWare)

129 following an emerging research topic in Earth system modeling (Di Baldassarre et al., 2015; Troy

130 et al., 2015) and water resources system analysis (Denaro et al., 2017; Giuliani et al., 2016; Khan

131 et al., 2017; Li et al., 2017; Mulligan et al., 2014) about coupled modeling approach. Utilizing BI

132 mapping in an ABM allows the agents' internal thinking processes and assocaited decision

133 uncertainty to be accommodated in the agent rules as well as explicitly represented in the causal

134 reasoning behind an agent's internal (psychological) decision-making (Kocabas and Dragicevic,

135 2013) while the CL model informs the agent's actions under changing socioeconomic conditions

136 (Murphy, 1976; Spiegelhalter et al., 1993). The San Juan River Basin in New Mexico, USA is

137 used as the demonstration basin for this effort. The calibrated BC-ABM-RiverWare model is used

138 to evaluate impacts of uncertain risk preception from all agents in this basin. In this study, multiple

139 comparative experiments of conventional rule-based ABM (i.e., without using the BL and CL) are

140 conducted to demonstrate the advantages of the proposed BC-ABM framework in modeling

141 human decision-making processes. We also evaluate the effect of changing external economic

142 conditions on an agent's decisions.

143   The paper is structured as follows. We introduce our methodology in Section 2. The

144 background of the case study area: the San Juan River Basin is presented in Section 3. We show

145 the calibration and different scenario results of the coupled BC-ABM-RiverWare model in Section

146 4 (Results). The institutional context as well as model limitation and future work are discussed in

147 Section 5 (Discussion) followed by the Conclusion Section.



## 2. Methodology

### 2.1. Develop a "two-way" coupled ABM-RiverWare model



River-routing and reservoir management modeling is designed to simulate the deliveries
of water within a regulated river system (Johnson, 2014). Many river-reservoir management
models have been developed to address different objectives within a geographic region such as
MODSIM, RiverWare, CALSIM (Draper et al., 2004), IQQM (Hameed and O'Neill, 2005), and
WEAP (Yates et al., 2005). These models use a "node-link" structure to represent the entire river
network where "nodes" are important natural (sources, lakes, and confluences) or human (water
infrastructures and water withdrawals) components and "links" represent river channel elements.
RiverWare, developed in 1986 by the University of Colorado Boulder, is a model of water
resource engineering system for operational scheduling and forecasting, planning, policy
evaluation, and other operational analysis and decision processes (Zagona et al., 2001). It couples
watershed and reach models that describe the physical hydrologic processes with routing and
reservoir management models that account for water use for water resources assessment.
RiverWare has a graphic user interface and uses an object-oriented framework to define every
node in the model as an "Object." Each object is assigned a unique set of attributes. These attributes
are captured as "Slots" in RiverWare. There are two basic types of slots: Time Series and Table
Slots for each Object to store either time series or characteristic data.  Details of RiverWare
structure and algorithm can be found at Zagona et al. (2001) and its website:
http://www.riverware.org/.
Coupling an ABM with a process-based model has been done before but mostly focused
on groundwater models such as Hu et al. (2017) and Mulligan et al. (2014). One of the few
examples that involve coupling with a surface water model, Khan et al. (2017) developed a simple



ABM that coupled with a physically-based hydrologic model, Soil and Water Assessment Tool.
In this paper, we perform a two-way coupling (data transfer back and forth between ABM and
RiverWare) between an ABM and RiverWare, where selected Objects in RiverWare are defined
as agents. To facilitate the two-way coupling, we utilize a convenient built-in tool within
RiverWare: the data management interface (DMI) utility which allows automatic data imports and
exports from/to any external data source (RiverWare Technical Documentation, 2017, see also
Figure S1).

**2.2. Quantify planned behavior with BI mapping and CL model**
The ABM developed in this paper, as an implementation of the TPB, consists of two
components: the Bayesian Inference (BI) mapping and the Cost-Loss (CL) modeling. This unique
setting allows us to explicitly describe human decision-making processes and asscociated
uncertainty casued by information ambiguity in water management decisions. We describe the
details in this section.
2.2.1.  The Bayesian Inference (BI) Mapping
In this study, the Bayesian Inference (BI) mapping is applied to specify a decision maker's
(or agent's) internal thinking processes by building a cognitive map (also called a causal structure)
between decisions (or taking a specific management behaviors) and relevant preceding factors that
could affect decision-making (Dorazio and Johnson, 2003; Pope and Gimblett, 2015). In this
setting, the goal of an agent is to develop a decision rule (or management strategy) that prescribes
management behaviors for each time step that are optimal with respect to its objective function.
The uncertainty associated with these management behaviors, arise from ambiguity, is specified
by a "risk perception" parameter (Baggett et al., 2006; Pahl-Wostl et al., 2008) representing the



extent to which decision-makers explicitly consider limited knowledge or belief about (future)
information in their decision-making process (Müller et al., 2013; Groeneveld et al., 2017). This
is the definition of Knightian uncertainty which comes from the economics literature where risk is
immeasurable or the probabilities are not known (Knight, 1921).

In the field of water resource management, a decision is often made based on whether the

preceding factor is larger (or less) than a prescribed threshold (i.e., exceedance). A simple example
is that a farmer' belief of changing the irrigation area will be affected by the forecast of water
stored in an upstream reservoir at the beginning of the growing season (i.e., water availability). In
this study, both the forecast of a certain preceding factor $f$ (a random variable) and an agent's
belief of taking a specific management behavior (or making a decision) $\theta$ can be represented as
probabilities shown in Equations (1) and (2):

$$P(f) = \frac{\text{\# of events that a preceding factor exceeds threshold}}{\text{\# of total events in modeling period}} \quad (1)$$

$$P(\theta) = \frac{\text{\# of events of taking a management action} (= make\ decision)}{\text{\# of total events in modeling period}} \quad (2)$$

The conditional probability as represented in Equation (3) describes the probability of a preceding
factor exceeding its threshold given a specific decision was made.

$$P(f|\theta) = \frac{P(f \cap \theta)}{P(\theta)} \quad (3)$$

The conditional probability obtained in Equation (3) is then used to calculate the joint probability
of both the preceding factor exceeding its threshold and a particular decision being made (Equation

4).

$$P(\theta \cap f) = P(f|\theta) \times P(\theta) \quad (4)$$

Alternatively, the joint probability can be computed with Equation (5).





$$P(f \cap \theta) = P(\theta|f) \times P(f) \tag{5}$$

Since the left-hand side of Equation (4) and (5) are mathematically equivalent, we can write their
right-hand side as

$$P(f|\theta) \times P(\theta) = P(\theta|f) \times P(f) \tag{6}$$

Rearranging Equation (6) provides a solution to $P(\theta|f)$ by Equation (7)

$$P(\theta|f) = \frac{P(f|\theta) \times P(\theta)}{P(f)} \tag{7}$$

The marginal probability can be written as:

$$P(f) = P(f \cap \theta) + P(f \cap \theta^c) \tag{8}$$

where $\theta^c$ means that the management behavior was not made. $P(f \cap \theta)$ is the probability of the
preceding factor exceeding its threshold when the decision was made, while $P(f \cap \theta^c)$ is the
probability of the preceding factor exceeding its threshold when the decision was not made.
Substituting Equation (8) into Equation (7):

$$P(\theta|f) = \frac{P(f|\theta) \times P(\theta)}{P(f \cap \theta) + P(f \cap \theta^c)} \tag{9}$$

Equation (9) can be rewritten by expanding $P(f \cap \theta)$ and $P(f \cap \theta^c)$,

$$P(\theta|f) = \frac{P(f|\theta) \times P(\theta)}{P(f|\theta)P(\theta) + P(f|\theta^c)P(\theta^c)} \tag{10}$$

where $P(\theta^c) = 1 - P(\theta)$ is the probability of not taking the management behavior $\theta$. In our case,
the information of $f$ is coming from RiverWare to ABM and $\theta$ is the result the ABM sends back
to RiverWare.
Equation (9) represents the probability of $\theta$ being made when the preceding factor exceeds the
given threshold. Similarily, $\theta$ being made when the preceding factor does not exceed the threshold
$(f^c)$ may be expressed as





$$P(\theta|f^c) = \frac{P(f^c|\theta) \times P(\theta)}{P(f^c|\theta)P(\theta) + P(f^c|\theta^c)P(\theta^c)} \tag{11}$$

The overall probability of taking a management behavior $P(\theta)$ relying on the preceding factor $f$
can be written using the law of total probability

$$P(\theta) = P(\theta|f) \times P(f) + P(\theta|f^c) \times P(f^c) \tag{12}$$

A solution of $P(\theta)$ can be obtained by substituting Equations (10) and (11) into (12)

$$P(\theta) = \frac{P(f|\theta) \times P(\theta)}{P(f|\theta)P(\theta) + P(f|\theta^c)P(\theta^c)} \times P(f) + \frac{P(f^c|\theta) \times P(\theta)}{P(f^c|\theta)P(\theta) + P(f^c|\theta^c)P(\theta^c)} \times P(f^c) \tag{13}$$

A general form of Equation (13) can be written as (Shafiee-Jood et al., 2017)

$$P(\theta) = \sum_i P(\theta|F_i) \times P(F_i) = \sum_i \frac{P(F_i|\theta)P(\theta)}{\sum_j P(F_i|\Theta_j)P(\Theta_j)} \times P(F_i) \tag{14}$$

where $F_i \in [f, f^c]$, $\Theta_j \in [\theta, \theta^c]$. In this study, we re-name the variables in Equation (13) as
follows

$$\begin{cases} \Gamma_{pr} = P(\theta) \\ \Gamma_{pd} = P(f) \\ \lambda = P(f|\theta) \end{cases} \tag{15}$$

where $\Gamma_{pr}$ represents the decision maker or agent's prior belief of $\theta$, $\Gamma_{pd}$ the probabilistic forecast
of preceding factor $f$, $\lambda$ the rate of acceptance of new information which represents a decision
maker's belief about the received information from $f$ (belief of the forecast/measurement accuracy
representing the degree of ambiguity of $f$).

By applying the BI theory to Equation (13) with the expressions in Equation (15), the

agent's prior belief of $\theta$, $\Gamma_{pr}^t$ at time $t$ can be expressed as

$$\Gamma_{pr}^t = \frac{\lambda \Gamma_{pr}^{t-1}}{\lambda \Gamma_{pr}^{t-1} + (1-\lambda)(1-\Gamma_{pr}^{t-1})} \Gamma_{pd}^t + \frac{(1-\lambda)\Gamma_{pr}^{t-1}}{(1-\lambda)\Gamma_{pr}^{t-1} + \lambda(1-\Gamma_{pr}^{t-1})} (1 - \Gamma_{pd}^t) \tag{16}$$

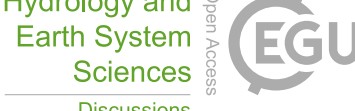



In Equation (16), the agent's prior belief of $\theta$ at timestep $t$, $\Gamma_{pr}^{t}$, is updated based on the prior belief
at previous timestep $t-1$, $\Gamma_{pr}^{t-1}$, and new incoming information or forecast at time $t$, $\Gamma_{pd}^{t}$. $\Gamma_{pr}^{t}$ lies
in between $\Gamma_{pr}^{t-1}$ and $\Gamma_{pd}$. Two extreme cases are described here. When $\lambda=1$, Equation (16)
reduces to $\Gamma_{pr}^{t}=\Gamma_{pd}^{t}$, which indicates that the agent's belief of taking management behavior is
purely based on the new incoming information, which corresponds to a risk-seeking decision
maker. In contrast, when $\lambda=0.5$, Equation (16) becomes $\Gamma_{pr}^{t}=\Gamma_{pr}^{t-1}$ suggesting that a decision
is made based on an agent's previous experiences alone (i.e., the decision maker's most recent
experience). This means that we have a risk-averse decision maker who totally ignores the new
incoming information (or no information arrived) and strictly makes his/her decision based on
his/her previous belief. In this study, the $\Gamma_{pr}^{t}$ in Equation (16) at each time step is updated by
applying the Bayesian probability theory to $\Gamma_{pr}$ between two consecutive time steps to take the
temporal causality between the two decisions into account.

In most water resources management cases, multiple preceding factors affect the

probability of a single management decision. In this paper, we assume that agents will make a
decision based on the most "highly recognized" preceding factor following the suggestion from
Sharma et al. (2013). The fundamental assumption is that a decision-maker will pay the closest
attention to the most abnormal of any preceding factors, such as the severity of droughts or floods,
historic low or high water levels of an upstream reservoir or an extreme upstream water diversion.
The way we represent this tendency is by calculating the "extremity" factors ($V$) of preceding
factors

$$V_i = \left| \frac{\theta_i}{\theta_{max}} - 0.5 \right| \qquad (17)$$





where $\theta_i$ is the $i^{\text{th}}$ preceding factor and $\theta_{max}$ is the maximal value of $\theta_i$. After the extremities of
all preceding factors have been calculated, agent will select the preceding factor with the highest
$V_i$ to update the prior belief of management actions based on Equations (16).
2.2.2.   The Cost-Loss (CL) Model
The BI mapping method described in Section 2.2.1 characterizes an agent's behavioral
intentions related to its internal (psychological) decision-making processes. According to the TPB,
a real-world management decision or action also depends on external uncontrollable factors such
as socioeconomic conditions. The CL model is applied in this study to address this concern. The
CL model measures the tendency of an adverse event affecting the decision of whether to take
costly precautionary action to protect oneself against losses from that event.  Based on the theory
of Cost-Benefit Analysis, if such event does not occur, the expected cost of taking action is "$C$"
and the expected loss of not taking action is "$L$". On the other hand, if such event does occur (with
a probability of $p$), the expected cost of taking action is still "$C$" and the expected loss ("$L$") of not
taking action is "$p \times L$." It follows that for one to take precautionary action, the expect cost of
taking that action should be less than the expected loss:

$$C \leq pL \tag{18}$$

Where Equation (18) can be rewritten as

$$z = \frac{C}{L} \leq p \tag{19}$$

where $z$ is defined as the Cost-Loss (CL) ratio and only when this value is less the probability of
the event occurring, the precautionary action will be taken.
To fit the CL model into the proposed ABM framework, we modify the above CL model
following the concept of Tena and Gómez (2008) and Matte et al. (2017) which added the




perception of risk into the decision-making process. We define "$C$" as the expected cost of taking
management action that will potentially increase the gross economic profit and "$L$" as the expected
opportunity loss of not taking such management action. The ratio of C to L (CL ratio $z$), as a
measure of tendency, can be compared with the prior belief of an agent's for taking a management
decision ($\Gamma_{pr}^{t}$ in Equation 16). When $\Gamma_{pr}^{t}$ is greater than $z$, this decision will become real world
management action since it makes economic senses.

$$\Gamma_{pr}^{t} \geq z = \frac{C}{L} = \frac{the\ expected\ cost\ of\ taking\ management\ action}{opportunity\ loss\ of\ not\ taking\ management\ action} \tag{20}$$

When $z$ increases, it means the cost of taking management action goes up or the opportunity loss
of not taking management action goes down. In either case, agents are less likely to take action
due to reduced profits. When $z$ decreases, following the same logic, agents are more likely to take
action.

Figure 1 summarizes the methodology in Section 2.2 applied to this study. Agent's

decision-making and action process will start when receiving information ($\Gamma_{pd}^{t}$) from RiverWare
and the conditional probability of its decision $\Gamma_{pr}^{t}$ will be computed after the most "highly
recognized" preceding factor is decided by the $V_i$ values. This probability of an agent's decision
will be compared with the CL ratio ($z$) to account for the external economic conditions where the
agent is located. The final management action from the agent will depend on whether the
probability of making a decision for an agent's is greater (take the action) or smaller (do not take
the action) than the CL ratio. This process is repeated annually throughout the entire simulation
period.



## 3. Case Study

**3.1. Study Area**

The San Juan River Basin (Figure 2) is the largest tributary of the Colorado River Basin with a drainage area of 64,570 km$^2$. Originating as snowmelt in the San Juan Mountains (part of the Rocky Mountains) of Colorado, the San Juan River flows 616 km through the deserts of northern New Mexico and southeastern Utah to join the Colorado River at Glen Canyon. Most water use activities are located in the upper part of the San Juan River Basin inside the States of New Mexico and Colorado. There are sixteen major irrigation ditches, four cities and two power plants (Figure 2) located in this basin and the water for which the San Juan River is the primary source. Major crops grown in the basin include hay, corn, and vegetables and the main planting season runs from May to October (Census of Agriculture – San Juan County, New Mexico, 2012). Navajo Reservoir, located 70 km upstream of the City of Farmington, NM is the main water infrastructure in the basin (Figure 2) which is used for flood control, irrigation, domestic/industrial water supply and environmental flows. The reservoir is designed and operated by the U.S. Bureau of Reclamation (USBR) following the rules in Colorado River Storage Project (Annual Operating Plan for Colorado River Reservoirs, 2017). The active storage of the reservoir is 1.3 million acre-ft (1.6 billion m$^3$). The maximum release rate is limited to 5000 cfs.

Beside the 16 major irrigation ditches, the Navajo Indian Irrigation Project (NIIP) is one another major water consumption within the basin that provides water to tribal communities in the region. San Juan-Chama Project manages transbasin water transfers into the Rio Grande Basin augmenting supply for Albuquerque, NM, irrigation and instream flow needs. Finally, the San Juan River Basin Recovery Implementation Program (SJRIP) implemented by the Fish and Wildlife Service, manages environmental flows within the basin, dictating timing and magnitude





of releases from Navajo Reservoir and maintainance of a daily 500 cfs minimum streamflow
requirement (Behery, 2017). Water rights within the San Juan River Basin in New Mexico have
not been completely adjudicated (see more details in Section 5.1). To address this gap, ten of the
largest water users have cooperated to develop a shortage sharing agreement to keep Navajo
Reservoir from drawing down the reservoir pool elevation below 5990 ft, which is the elevation
required for NIIP diversion. The agreement stipulates that all parties share equally in shortages
caused by drought (2013-2016 shortage agreement is available at: https://www.fws.gov/-
southwest/sjrip/DR_SS03.cfm).
**3.2. The Coupled ABM-RiverWare Model Setup**
USBR developed a RiverWare model for the San Juan River Basin to support water
management and resource planning efforts. RiverWare includes 19 irrigation ditches objects, 21
domestic and industrial use objects, two power plant objects and three reservoir objects. Input data
for the RiverWare model include historical tributary inflows, evapotranspiration rates for each
irrigation ditches limited by the crop water requirement, historic water diversion for NIIP and the
San Juan-Chama Project, and reservoir operations rules. Ungaged local inflows were determined
by the simple closure of the local water budget. The model operates on a daily time step from
10/01/1928 to 09/30/2013 (85 years) with four "cycles" of simulation. Each cycle is a complete
model run for the entire modeling period to fulfill part of the necessary information (e.g., some
downstream water requirements need to be pre-calculated for Navajo Reservoir to set up the
release pattern). Shortage sharing is handled at Cycle 3 and Cycle 4 of the model run. In Cycle 3,
if the water level in the Navajo Reservoir at the end of a water year (23:59:59 at September 30[th])
is lower than the threshold (5990 ft above sea level) , RiverWare will mark the coming year as a
"water shortage year."   Then, in Cycle 4, shortage sharing rules dictate Navajo Reservoir





Operations and water deliveries to basin water users. Effectively, Cycles 1 to 3 determine the
volume of water available for use such that environmental flows are achieved and Navajo
Reservoir levels are maintained above 5990 ft. Where the available supplies are less than the
desired use, all users share equally in the shortage.
In this study, the 16 major irrigation ditch objects in RiverWare are defined as agents. At
the end of every water year, each of the 16 agents decided whether to expand or reduce their
irrigated area for the coming year. We categorized the 16 agents into three groups based on their
location (colored in Figure 2). Agents in Group 1 (light blue) were located upstream of the Navajo
Reservoir; Group 2 (light green) were located on the Animas River (a major tributary of the San
Juan River), and Group 3 (orange) were located downstream of the Navajo Reservoir.
The BI mapping was applied to each group with different causal structures. The preceding
factors that affected Group 1 agents' decisions are: (Navajo) upstream winter precipitation, the
water level in Navajo Reservoir and flow violations at the basin outlet (days below 500 cfs in a
water year). Group 2 agents consider local (Animas River) winter precipitation, upstream winter
precipitation, the water level in Navajo Reservoir and flow violations at the basin outlet. Group 3
agents consider local (downstream of Navajo Reservoir) winter precipitation, upstream winter
precipitation, the water level in Navajo Reservoir, flow violations at the basin outlet, and NIIP
diversions. The agents' information is listed in Table 1. In this study, flow violation at the basin
outlet and water level of Navajo Reservoir are two representative factors of social normative
pressures as it is considered by all the three groups of agents. In contrast, other factors reflect the
diverse internal thinking processes among agent groups due to their geographical locations. Note
that the information of winter precipitation was not taken from RiverWare; rather, was gathered
from NOAA ground-based rainfall monitoring gauges where we used the coming year's winter





precipitation as a proxy for the snowpack forecast in the causal structure. Agents that participated
in the shortage-sharing plan are also considered if a shortage had been declared. If a shortage were
declared, the RiverWare model would reduce the targeted streamflow at the basin outlet to 250 cfs.
The participating six agents adjust their water diversion to achieve this newly targeted streamflow
under these extreme drought conditions. Under the current setting, agents follow the "backward-
looking, forward-acting" mode, which means that agents make decisions based on their own
past/current experiences (water level in Navajo Reservoir, stream flow violations at the basin outlet,
NIIP water diversion, and the previous decision on the irrigated area) and their belief of the winter
precipitation forecast in the coming year.

The detailed causal structure of BI mapping for each type of agent are given in the

supplemental material where a standard "Overview, Design concepts, and Details" (ODD)
protocol for ABM development is followed as suggested by Grimm et al. (2010). Finally, the data
transfer from RiverWare to ABM at the end of a water year included 1) irrigation areas for the 16
irrigation agents, 2) the basin outflow, 3) water level in the Navajo Reservoir and 4) the NIIP water
diversion. Following the ABM simultaion, data transfer from ABM to RiverWare included 1)
updated irrigated areas and 2) the corresponding water diversion of each agent.
**3.3. ABM-RiverWare Model Calibration**

One of the major criticisms of ABM development is that ABM results are difficult to verify

or validate (Parker et al., 2003; An et al., 2005, 2014; National Research Council, 2014). In this
study, we address this concern by calibrating the coupled BC-ABM-RiverWare model to match
the historical patterns of 1) individual agent's irrigated area and 2) San Juan River discharge.
USBR provides the observed irrigated acreage for all 16 ditches and the flow measurements at two



different locations along the San Juan River (at the outlet of the San Juan River Basin and directly
downstream of the Navajo Reservoir) for calibration purpose.

The calibrated parameters related to an agent's decision-making processes are the risk

perception parameters ($\lambda$) and CL ratio ($z$) of each individual agent. In this study, each agent has
four $\lambda$s characterized by the relative geographical location with considered preceding factors.
Unique values of $\lambda$ are assigned to each preceding factor for each agent (through calibration) as
different agents should have different levels of risk tolerance for each preceding factor. Different
values of $z$ are assigned to each agent representing the spatial heterogeneity of socioeconomic
conditions. $z$ is assumed to be constant for each agent throughout the model period as relative up-
front cost information is unavailable. We also calibrate the irrigated areal increment of each agent
to quantify the capability of different farmers for expanding or reducing their farmland. The actual
irrigation area change at each year for each farmer is specified by the calibrated irrigated areal
increment with an added uncertainty of 30% representing the execution uncertainty of farmers.
The thresholds of each preceding factor are also calibrated for calculating the extremities. A total
of 102 parameters (Table 1) are manual calibrated ("trial-and-error") for this specific case in this
study and further explained in the supplement materials. In general, we calibrated the parameters
sequentially from upstream and tributary agents (i.e. Groups 1 and 2) to downstream (i.e. Group
3). Within a group, we calibrated agents with larger irrigated area first to capture a better system-
wide result.
## 4. Result
**4.1. BC-ABM-RiverWare Model Diagnostics**



The BC-ABM calibration results for individual agent's irrigated area from 1928 to 2013
are given in Figure 3 and arranged by group (region). The blue curves are the historical irrigated
area. The red curves are the best-fit result among multiple (30) modeling runs (shown by the gray
curves, which represents the stochasticity) of each agent. In general, BC-ABM captures the pattern
and trend of irrigated area for all agents, and we particularly focus on agents with the largest
irrigated areas since their decision can dominate the basin. A figure showing the time variations of
extremity values for each group of agents is given in the Supplementary Materials (see Figure S2)
to illustrate the preceding factors adopted by different groups of agents for making decision at each
time step.
The overall (area) weighted Nash-Sutcliffe Efficiency (NSE) of the best-fit result is 0.55
which represents a reasonable calibration result. There are a few cases where structural changes
occurred with some of the ditches that the model does not capture. Specifically, construction of
Navajo Reservoir in the 1960 inundated the New Mexico Pine River Ditch, while construction of
the dam made it possible to expand the Hammond Irrigation Ditch (located directly downstream
of Navajo Reservoir). Similar structural changes are evident with the Echo, New Mexico Animas
and Fruitland-Cambridge Ditches. The model limitation associated with the use of BI mapping in
ABM is discussed in the Discussion Section.
To demonstrate the enhanced performance of the proposed BC-ABM framework in
representing human decision-making processes, we conducted comparative experiments with
conventional rule-based ABMs, which exclude the BI mapping and CL ratio, referred to as the
Non-BC-ABMs. In the Non-BC-ABMs, agents make decision based on either past experience (e.g.,
flow violation or NIIP diversion) or future forecast (winter precipitation) alone implying that
agents have a perfect foresight in received information. Using precipitation as an example, an agent



will expand irrigation area if the precipitation forecast is greater than the given threshold, and vice
versa. Excluding BI mapping implies that the agents make decision purely based on the forecast
or new incoming information and fully ignore the information from past experience, while
excluding CL model means that the agents do not take socioeconomic factors into account when
making decisions.

Multiple Non-BC-ABMs, in terms of considering different preceding factors and

including/excluding extremity in decision-making processes, were tested and results are also
shown in Figure 3. The black solid curve represents the Non-BC-ABM simulation utilizing
extremity for selecting the reference preceding factor, while the black dashed curve is the Non-
BC-ABM using only the precipitation in the decision-making processes. The better performance
of the proposed BC-ABM framework, compared to the Non-BC-ABMs, is evidenced by the closer
agreements between the simulated and historical patterns of irrigated area from BC-ABM as well
as weighted NSE (0.55 for BC-ABM vs. -1.25 for the Non-BC-ABM with extremity and -1.39 for
the Non-BC-ABM with precipitation alone). Different Non-BC-ABM simulations are also
compared with the BC-ABM simulations as shown in Figure S3. The time variations of $\Gamma_{pr}^{t}$ and
calibrated $z$ for each agent are shown in Figure 4 to illustrate the characteristics of different agents,
in terms of risk perception. The results show that the agents in Group 1 and 2 have a consistently
lower willingness to adjust irrigation area ($\Gamma_{pr}$ shown in red) compared to the corresponding CL
ratio ($z$ shown in black). In contrast, Group 3 agents adjust irrigation area more often as evidenced
by the frequent crossover between red and black curves, which suggest that agents in Group 3 are
relatively risk-neutral compared to those in Group 1 and 2.

The calibration results of basin outflow and Navajo Reservoir inflow from 1928 to 2013

are given in Figure 5. The results show that the simulated values of both quantities agree closely



with the historical observations evidenced by the NSEs of 0.60 and 0.54, respectively. The multi-
objective calibration conducted in this study can capture not only human agents' activities but also
the basin level water balance. On the other hand, multi-objective calibration also reduces the
potential equifinality (Beven, 2006) resulting from a large number of model parameters and a
limited number of observations, especially for a high-dimensional complex system such as CHNS
(e.g., Franks et al., 1998; Kuczera and Mroczkowski, 1998; Yapo et al., 1998; Choi and Beven,
2007). We do notice that our coupled BC-ABM-RiverWare simulation misses peaks of streamflow
possibly due to the lower input data of RiverWare. However, since our focus is the water-scarce
situation in this case study, this underestimation does not significantly affect our following analysis.
**4.2. The effect of agents' risk perception**

The calibration results in Section 4.1 demonstrate the creditability of the coupled BC-

ABM-RiverWare model in representing human psychological, uncertain decision-making process.
The encouraging results suggest that we can apply the proposed BC-ABM framework to test some
"extreme conditions" to perform different scenario analyses. Different scenarios in terms of risk
perception were tested by making stepwise change of all $\lambda$ values from "0.5" (risk-averse) to "1"
(risk-seeking). The basin-wide results are summarized in Figure 6 which shows the key measure
quantities including cumulative probability distribution of annual total irrigated area, Navajo
Reservoir water level in December, annual total downstream flow violation frequency and volume.
The simulations under extreme risk-averse ($\lambda = 0.5$) and risk-seeking ($\lambda = 1$) scenarios are shown
in blue and green, while those with calibrated historical risk perceptions for each agent are shown
in red, referred to as the baseline simulation. The gray curves lying between blue and green are the
estimates of these measured quantities corresponding to different $\lambda$ values. The total irrigation area



generally increases with an increase in $\lambda$, indicating that the agents become more risk-seeking if
they are more confident about new incoming information.

There are two interesting observations. First, it is clear that when all agents become risk-

seeking, their actions become more aggressive and result in greater irrigated area in the basin
(Figure 6a). Since all agents want to expand their irrigated area, Navajo Reservoir will reserve
more water at the end of each year resulting in slightly higher water levels (Figure 6b). Streamflow
violations show a somewhat counterintuitive result. The volume of violation under risk-seeking
scenario increases as expected (green curve shifts to right in Figure 6d) but the frequency of
violation decreases (green curve shifts to left in Figure 6c). This is due to that Navajo Reservoir
holds more water for irrigation season to satisfy downstream increased water demand which will
result in much fewer streamflow violation days during the irrigation season. Although this
operation slightly increases streamflow violation days in the following season, the total violation
days still decrease (Figure S4 in the Supplementary Materials). Second, the results of baseline
simulation (red curves) are very close to the "all agents risk-averse" scenario results (blue curves).
This is consistent with the findings from previous studies (e.g., Tena and Gómez, 2008), which
suggest that farmers are commonly risk-averse when the stakes are high (Matte et al., 2017).

We also look at the time variations of individual irrigated area changes for characterizing

risk perceptions of different agents. Figure 7 shows the simulated irrigation area changes for four
selected large irrigated areas since they are the major "players" in the basin. It again characterizes
different agents' behavior (see also Figure 3). The results clearly show that Jicarilla (Group 1) and
NMAnimas (Group 2) are historically risk-averse agents. In contrast, Hammond and Hogback
(Group 3) are relatively risk- neutral, compared to agents in Group 1 and 2, as the red curves lie in
between green and blue curves. Group 3 agents are located downstream of Navajo Reservoir, and



their decision-making process considers the water level in the reservoir (reflected by their BI
mapping). Most of Group 3 agents consider Navajo Reservoir as a steady water source so they can
have relatively more aggressive attitudes toward risk compared to their upstream counterparts.
Also, note that Jicarilla, Hammond, and Hogback under the risk-seeking scenario eventually reach
their maximum available irrigated area. Therefore, their irrigated area flattens out at the end of the
simulation. The gray curves in Figure 7 represent the simulated irrigation area changes for agents
corresponding to different agents' risk perceptions. It shows that the irrigation area generally
increases with an increase in $\lambda$ for all the four agents.

**4.3. The effect of socioeconomic condition**

The proposed BC-ABM framework allows us to quantify the influences of external
socioeconomic factors on human decision-making processes by adjusting the CL ratio. In this
study, we conducted a sensitivity analysis on the cost-loss ratio to test "*what if economic conditions*
*change and it becomes more expensive or cheaper to expand the irrigated area*" by systematically
increasing (+10% and +20%) or decreasing (-10% and -20%) all z values. When the z value goes
up, it means that the cost of increasing irrigated area goes up, or the opportunity loss of not
increasing irrigated area goes down. In either case, the situation should become harder for agents
to expand their irrigated area. When the z value goes down, following the same logic, the economic
conditions become easier for agents to expand their irrigated area. The modeling results shown in
Figure 8 fit with this intuition quite well. All blue and cyan curves (increasing z values) are located
below, and purple and magenta curves (decreasing z values) are located above red curves (baseline
simulations). Modeling results also show that in the basin, Groups 1 and 2 are less sensitive to the
changes in economic conditions but agents in Group 3 are more sensitive to the economic
conditions. Of course, individual differences exist inside each group.



According to the San Juan River Basin regional water plan, several strategies and
constructions such as on-farm and canal improvements and municipal and irrigation pipeline from
Navajo Reservoir, will be authorized for meeting future water demand (State of New Mexico
Interstate Stream Commission, 2016). These strategies and constructions could lead to a change
of future socioeconomic conditions, in terms of the cost of water usage and changing irrigated area
(e.g., up-front cost) for stakeholders. In this study, we quantify the effects of up-front cost on the
changes of irrigation area (i.e., irrigation water demand) using the proposed BC-ABM framework.
According to the proposed ABM framework, up-front cost could affect human decision-making
processes from two perspectives. First, it directly changes the socioeconomic condition of an agent
(change of CL ratio). Second, it influences an agent's decision-making processes by introducing
more information (change of causal network in BI mapping). As a result, the proposed BC-ABM
framework can take up-front costs into account without theoretical and technical difficulties if
related information is available. Two scenarios assuming a general increasing and decreasing up-
front cost for agents over time, are tested in the study, respectively. For each agent, a time varied
$z$ is generated by adding a positive/negative trend with a small random fluctuation to the calibrated
$z$ to mimic the spatial and temporal heterogeneity of up-front costs. Note that we did not include
up-front cost into current BI mapping as it requires real data from all stockholders to re-calibrate
all the model parameters.
The time variation of irrigated area for all 16 agents under different economic scenarios
are shown in Figure 9. The cyan and green curves are the irrigated area change under an increasing
and decreasing z, respectively, while red curves are the simulations from baseline case using
calibrated z values. The results show that Group 1 and Group 2 agents are not affected by the
changing z significantly. The influence of changing z on Group 3 agents is relatively significant.



A consistently higher (lower) green (cyan) curve as compared to the baseline simulation is found.
The preliminary results are expected as they fit the economic intuition. In this specific case,
farmers tend to expand their irrigation area earlier (by comparing cyan and red curves) if they
expect a corresponding increased cost in the future. In contrast, if the cost of expanding irrigation
area in the future is expected to go down, farmers will defer the actions to pursue a lower cost.
## 5. Discussion
**5.1 Water policy implementation in the San Juan River Basin**

The method proposed in this paper is intended to be a generalizable approach to explicitly

characterize human decision-making processes and quantify the associated uncertainty due to
information ambiguity in watershed management. The real-world decisions regarding irrigated
area change and water management are often more complex than the proposed BC-ABM,
especially for a watershed like San Juan River Basin that has a complex institutional context. To
illustrate the potential application and broaden the impact of this case study, we summary the
policy implementation in the San Juan River Basin in this section.

To improve water planning and management in San Juan River Basin, a steering committee

constitute of several state and federal agency representatives was established with the main
responsibility of overseeing the institutional complexity for the water plans operated under the
1922 Colorado River Compact and 1948 Upper Colorado River Basin Compact. Although a
regional water plan report (RWP) was updated in 2016 (State of New Mexico Interstate Stream
Commission, 2016) by interested stakeholders, issues still exist under the terms of 1948 Upper
Colorado River Basin Compact. For example, New Mexico's entitled 642,380 acre ft. consumptive
use is substantially greater than the corresponding consumptive use.



The RWP summarizes the related information of water planning such as water rights, future
water supply and demand projections, and newly available data. The analysis states that the total
water demand in the San Juan Basin is expected to increase due to the authorized expansion of
NIIP irrigation area, while a reduction of future water supply due to climate change is anticipated
based on the regional assessments conducted by the U.S. Global Change Research Program. A San
Juan Navajo Water Rights Settlement was executed in December 2010 for confirming the
provisions of the related water supply contract for the Navajo Nation. Even so, several pending
adjudications still exist. For example, the current water rights settlements are based upon existing
irrigation projects, which may potentially displace the existing non-Navajo water uses.
Additionally, part of water supply information is less reliable (e.g., tributary diversion). The RWP
also identified several key issues (e.g., stream restoration, water quality protection, irrigation
conveyance efficiencies, water banking, and land use.) and strategies (e.g., water system
infrastructure upgrade and improvement) for the improvement of water resource management
within the region.
Since irrigation activities are the most consumptive components of water demand among
others, (74.8% of total water demand, State of New Mexico Interstate Stream Commission, 2016),
collective adaptive actions of farmers will significantly affect the water planning and management
in San Juan by e.g., changing the water diversion and reservoir release. The BC-ABM results
presented in Section 4 have shown that farmers react to changing climatic and socioeconomic
conditions. Understanding and accounting for the adaptive capacity of regional water resources in
response to farmer's behaviors is critical to the management of scarce water resources. For
example, the sensitivity analysis (see Figure 8) suggests that the collective action of farmers has
potential to influence the irrigation of $4.5 \times 10^4$ to $6.1 \times 10^4$ acres of cropland with 9000 to 12000 ac-



ft of water demand, which is about 30 to 39% of average annual water usage under changing
economic conditions (i.e., 20% increase or decrease of up-front cost). A potential
increase/decrease of future irrigation cost could also influence farmers' decisions. Understanding
such behavior is also critical to future water resource planning and management in the San Juan as
(1) threat of climate change will lead to shift in timing of flows associated with a mean decrease
in future water supply resulting from an anticipated reduced precipitation and/or increased
evaporation, and (2) there are several settlement agreements with the tribal communities along the
San Juan where their committed allotment of water has yet to be put to full use (e.g., Navajo Gallup
Pipeline and Navajo Indian Irrigation Project that both require construction and/or expansion of
existing water delivery infrastructure to make full use of water rights).
**5.2 Limitation and future study**

Here we discuss several limits or aspects that we did not fully cover in this paper and

potential future research directions. First, we focus on the methodologies of model development
(i.e., parameter calibration of BC-ABM) specifically, rather than the precise causal structure of BI
mapping (Cheng et al., 2002; Premchaiswadi et al., 2010). We aim to demonstrate that the
proposed BC-ABM framework can effectively capture agent's risk perception. In general, an
accurate causal structure of BI mapping can be obtained by a detailed interview with decision
makers (O'Keeffe et al., 2016) or learned from a dataset (Sutheebanjard and Premchaiswadi, 2010).
Missing information of factors associated with individual decision making could impact the
calibration reuslts. A typical example is that our model does not capture the abrupt change of
historical irrigation area change (e.g., Hammond in Figure 3) caused by the missing of key factors
in addition to climate conditions (Navajo Reservoir came online implying a change in the irrigation
system).



Second, the external socioeconomic condition ($z$) for each agent is treated as a calibrated
parameter in current ABM framework. The value of $z$ can be estimated directly by acquiring
relevant data, if available. For example, the farm production expense data provided by U.S.
Department of Agriculture could be used as an approximation of the expected cost of changing
irrigation area ($C$ in Equation 20), while the farm income and wealth statistics estimated from crop
production may be considered as a major part of opportunity loss ($L$ in Equation 20). Third, the
up-front cost is not included in the current BC-ABM framework and performed analysis due to
lack of information as mentioned in Section 3.3 and 5.1. A potential solution, upon further tests,
is to include up-front cost in the current BI mapping and add the up-front cost to CL ratio whenever
it appears as the preceding test presented in Section 5.1.
Fourth, other than the extremity used in this study to be the reference of agent's decision,
techniques and methods for multi-criteria decision analysis such as the Analytical Hierarchy
Process (AHP, Saaty and Vargas, 2001) or Analytic Network Process (ANP, Saaty and Vargas,
2006), also has potential to be incorporated into current ABM framework as a tool of integrating
multiple source of information. These methods are usually served as decision-support tools by
evaluating the relative merit among different alternatives (e.g., preceding factors in this study).
Finally, the current method does not explicitly consider direct interaction among agents. We do
model agents as interacting indirectly through irrigated water withdrawal (i.e., upstream agents'
water uses will affect downstream agents' water availability). However, effects like "peer-
pressure," "word-of-mouth" and potential water markets are not currently considered in the model.
Future work is planned toward methodology development to include direct agent interaction into
the BI mapping. Agents' decisions can be affected by observing its neighbor's actions, and this



information will always be treated with $\lambda = 1$. This means agents will always believe their own
observations (i.e. "to see is to believe").

## 6. Conclusion

Managing water resources in a complex adaptive natural-human system subject to

uncertainty is a challenging issue. The interaction between human and natural systems needs to be
modeled explicitly with associated uncertainties characterized and managed in a formal manner.
This study applies a "two-way" coupled agent-based model (ABM) with a River-Reservoir
management model (RiverWare) to address the interaction between human and natural systems
using Bayesian Inference (BI) mapping joined with Cost-Loss (CL). The advantage of ABM is
that by defining different agents, various human activities can be represented explicitly while the
coupled water system provides a solid basis to simulate the environment where these agents are
located.

Joining BI mapping and CL modeling has allowed us to 1) explicitly describe human

decision-making processes, 2) quantify the associated decision uncertainty caused by
incomplete/ambiguous information, and 3) examine the adaptive water management in response
to changing natural environment as well as socioeconomic conditions, which extends previous
research where treatment of uncertainty has been largely limited to the natural system alone.
Calibration results for this coupled ABM-RiverWare model, as demonstrated for the San Juan
River Basin, show that this methodology can capture the historical pattern of both human activities
(irrigated area changes) and natural dynamics (streamflow changes) while quantifying the risk
perception of each agent via risk perception parameters ($\lambda$). The scenario results show that the
majority of agents in the basin are risk-averse which confirm the conclusion of Tena and Gómez



(2008). The improved representation of the proposed BC-ABM is evidenced by the closer
agreement of BC-ABM simulations against observations, compared to those from an ABM without
using BI mapping and CL ratio. Changing economic conditions also yield intuitive agent behavior,
that is, when crop area expansion is more expensive/cheaper, fewer/more agents will do it.
The current method does not focus on obtaining an accurate causal structure of the BI
mapping, which can be improved via survey or interview with local decision makers. Up-front
cost can be incorporated in current ABM framework by modifying the causal network of BI
mapping and adjusting the CL ratio whenever up-front cost appears. Considering different types
of agents and addressing the direct agents' interactions are other two future research directions.
**7. Acknowledgement**
This research was supported by the Office of Science of the U.S. Department of Energy as
part of the Integrated Assessment Research Program. Special thanks are given to Ms. Susan Behery
in USBR for providing historical data in the San Juan Basin. And Mr. Majid Shafiee-Jood in the
University of Illinois who discussed the methods of BI mapping and CL modeling with us in the
earlier version of the draft. All data used for both the RiverWare model (inflow, crop ET, and
water diversion, etc.) and the Agent-based Model (winter precipitation, historical basin outflow,
and historical irrigated area, etc.) are explicitly cited in the reference list.

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





## Tables

Table 1. Name of agent groups, number of agents in each group and the factors considered in decision-making processes for each group of agents in this study

| Group | Number of agents | Factors considered in decision-making processes |
|---|---|---|
| **1.** (upstream of the Navajo Reservoir) | 2 | Upstream Precipitation, Water Level in the Navajo Reservoir, Flow Violation at the outlet, Cost-loss ratio |
| **2.a** (Animas River without shortage sharing) | 5 | Animas Precipitation, Upstream Precipitation, Water Level in the Navajo Reservoir, Flow Violation at the outlet, Cost-loss ratio |
| **2.b** (Animas River with shortage sharing) | 1 | Animas Precipitation, Upstream Precipitation, Water Level in the Navajo Reservoir, Flow Violation at the outlet, Shortage Sharing, Cost-loss ratio |
| **3.a** (downstream of the Navajo Reservoir without shortage sharing) | 3 | Downstream Precipitation, Upstream Precipitation, Water Level in the Navajo Reservoir, Flow Violation at the outlet, NIIP Diversion, Cost-loss ratio |
| **3.b** (downstream of the Navajo Reservoir without shortage sharing) | 5 | Downstream Precipitation, Upstream Precipitation, Water Level in the Navajo Reservoir, Flow Violation at the outlet, NIIP Diversion, Shortage Sharing, Cost-loss ratio |





**Figures**

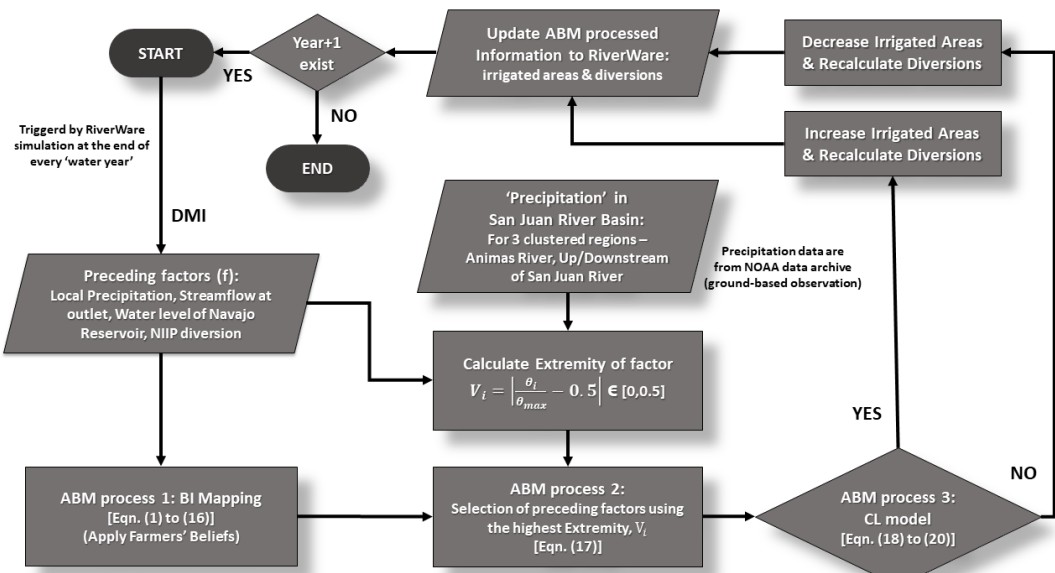

Figure 1. The flow chart of agent decision-making process inside the two-way coupled ABM-
RiverWare model (ABM.exe in Figure S1). Agents make their decisions with uncertainty
based on the method developed by this paper (joint BI mapping and CL model), and
RiverWare will run the simulation based on these decisions.



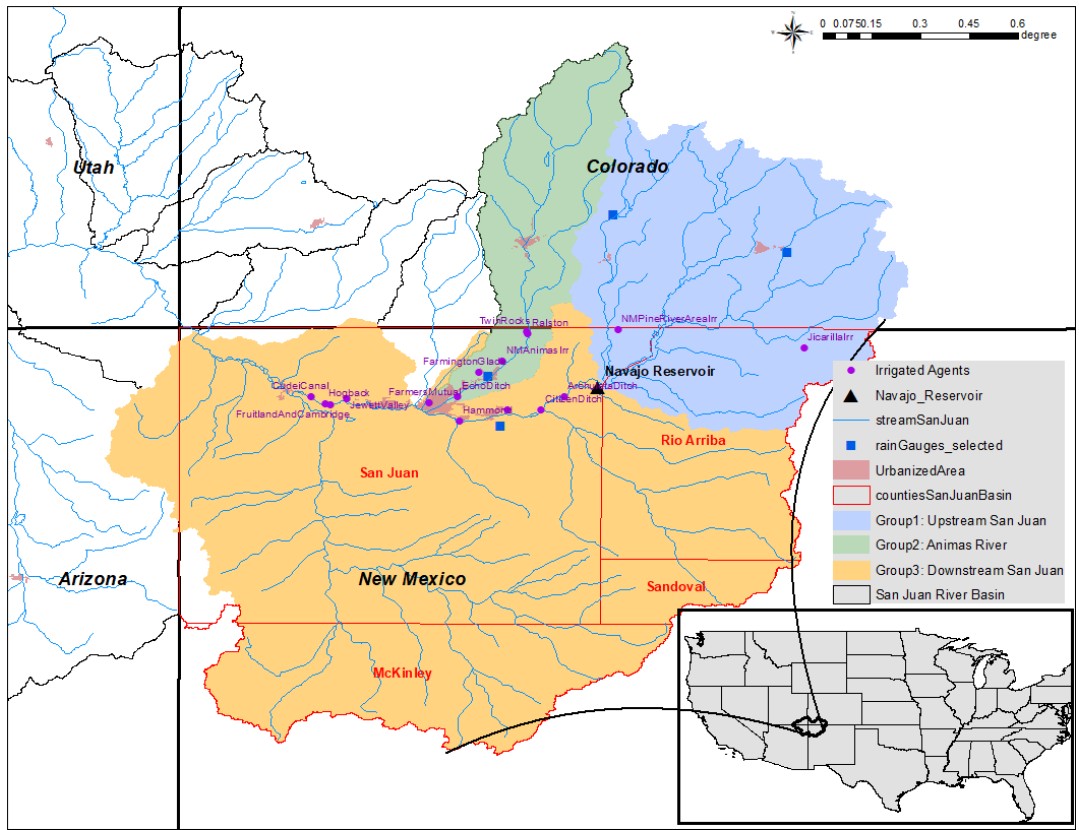

Figure 2. The upper San Juan River Basin. Different colors of the basin represent the geographical
regions that this paper used to group major irrigation districts (agents, marked as dots).
The location of Navajo Reservoir is marked as a triangle.





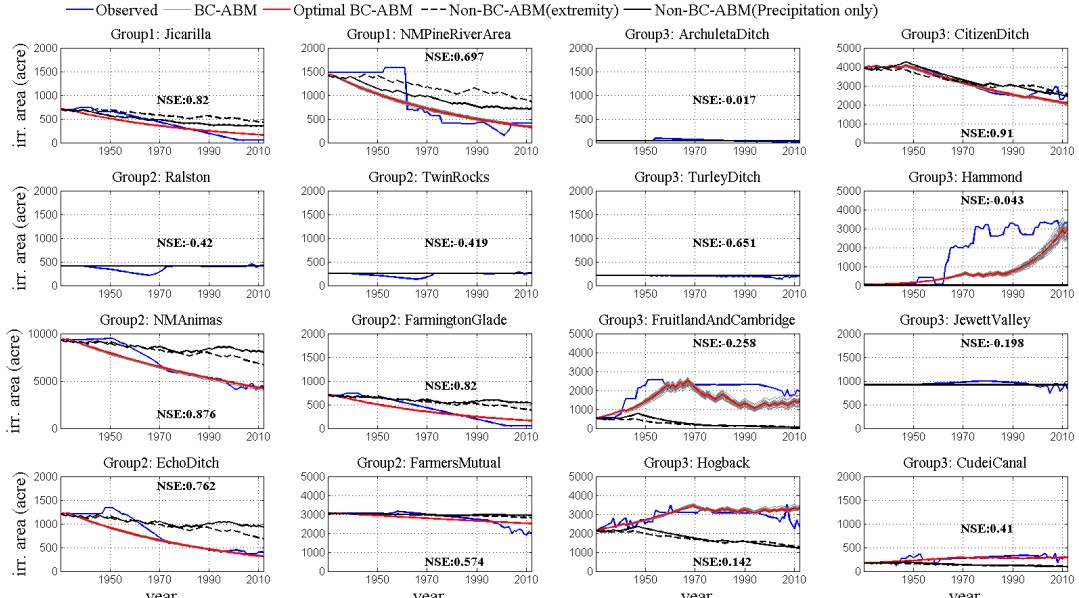

Figure 3. The calibration results of the ABM-RiverWare model: Individual agents' irrigated area changes from 1928 to 2013 organized by irrigation ditch and region (see groups in Figure 3). Each figure includes the simulated irrigated area change from the best-fit BC-ABM (solid red) and the corresponding Nash-Sutcliffe Efficiency (NSE), multiple runs of BC-ABM (solid gray) to visualize the stochasticity (30 runs) of agents' random behavior, Non-BC-ABM with extremity (dashed black), Non-BC-ABM using precipitation only (solid black) against historical record (solid blue).





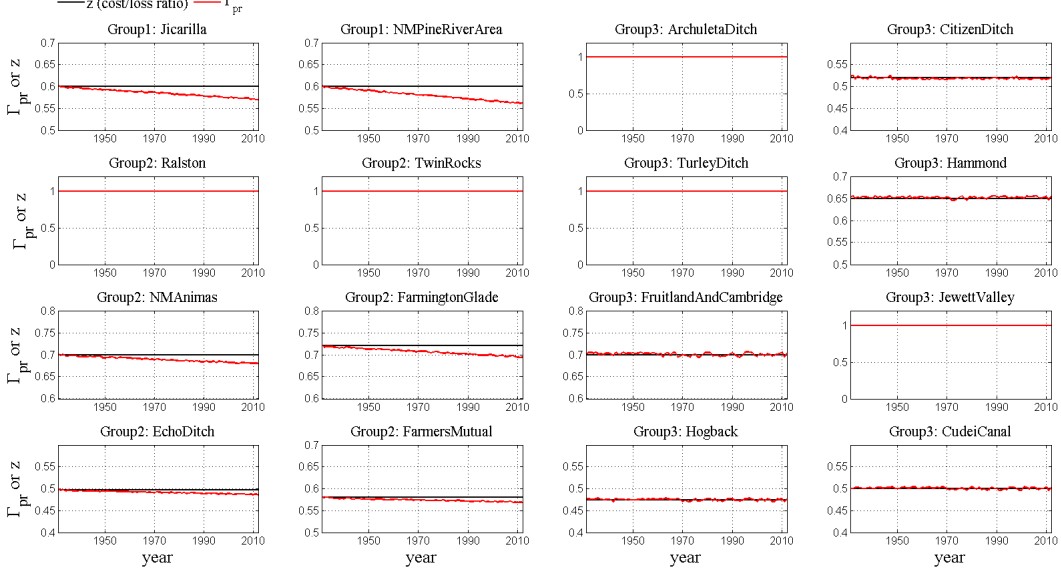


Figure 4. Calibrated probability of expanding area ($\Gamma_{pr}$) and cost-loss ratio ($z$) for each agent



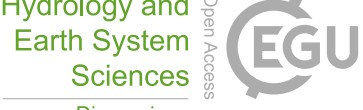

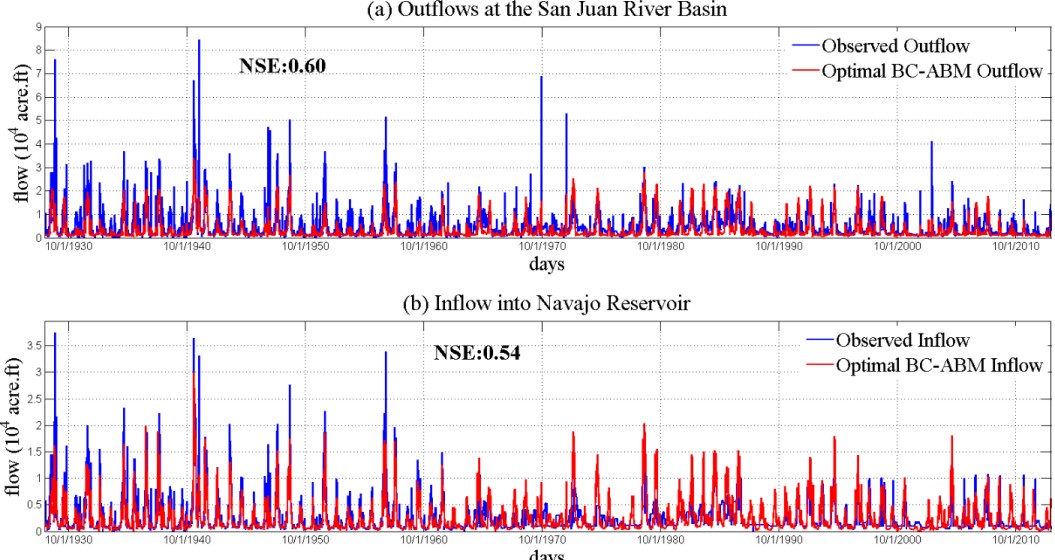

Figure 5. The calibration results of the ABM-RiverWare model: (a) the basin outflow to Colorado
River; (b) inflow to Navajo Reservoir. Blue lines are historical data and red lines are
modeling results.



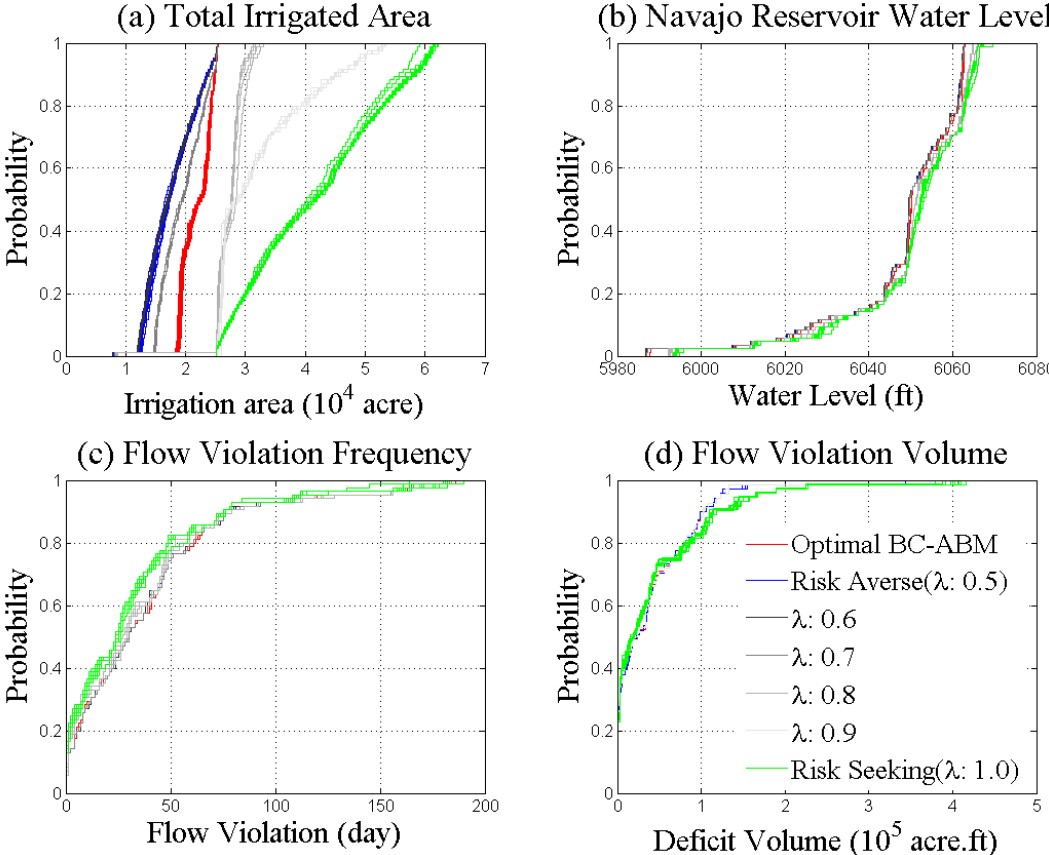

Figure 6. The cumulative density frequency throughout the entire simulation period of (a) basin-
wide irrigated area; (b) Navajo Reservoir end of the year water level; (c) basin outlet
annual streamflow violation days; (d) basin outlet annual streamflow violation volume.
Results are given for the calibrated (green curves), risk-averse (blue curves) and risk-
seeking (red curves) cases. The simulation results with different values of agents' risk
perceptions (λ) between 0.5 and 1 are shown in gray.



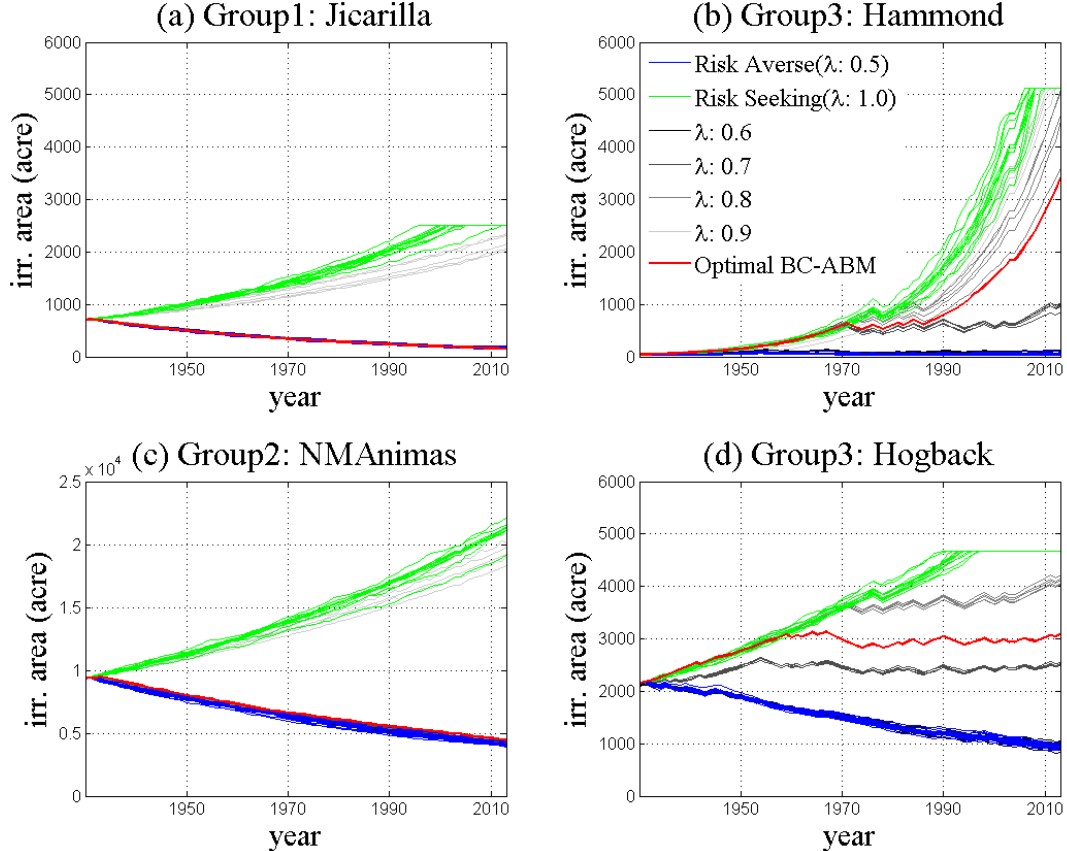

Figure 7. Individual agents' irrigated area changes under calibrated (green curves), risk-averse
(blue curves) and risk-seeking (red curves) scenarios. The simulation results with
different values of agents' risk perceptions ($\lambda$) between 0.5 and 1 are shown in gray.





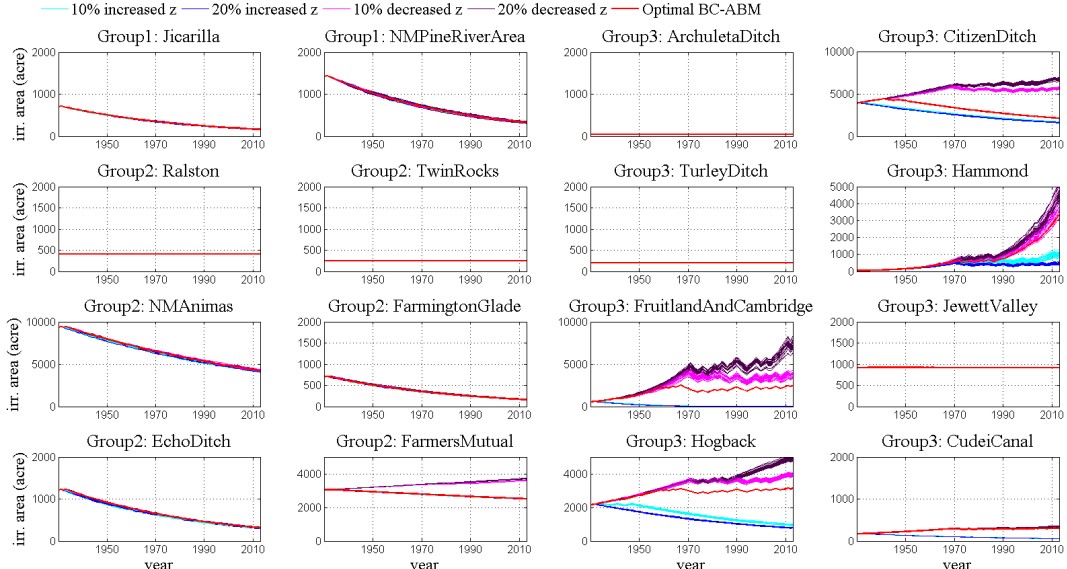

Figure 8. The sensitivity analysis of changing economic conditions on an agent's decision on
irrigated areas. Blue (+20%) and cyan (+10%) curves represent increasing z values which
make area expansion more expensive. Purple (-20%) and magenta (-10%) lines represent
decreasing z values which make area expansion cheaper.





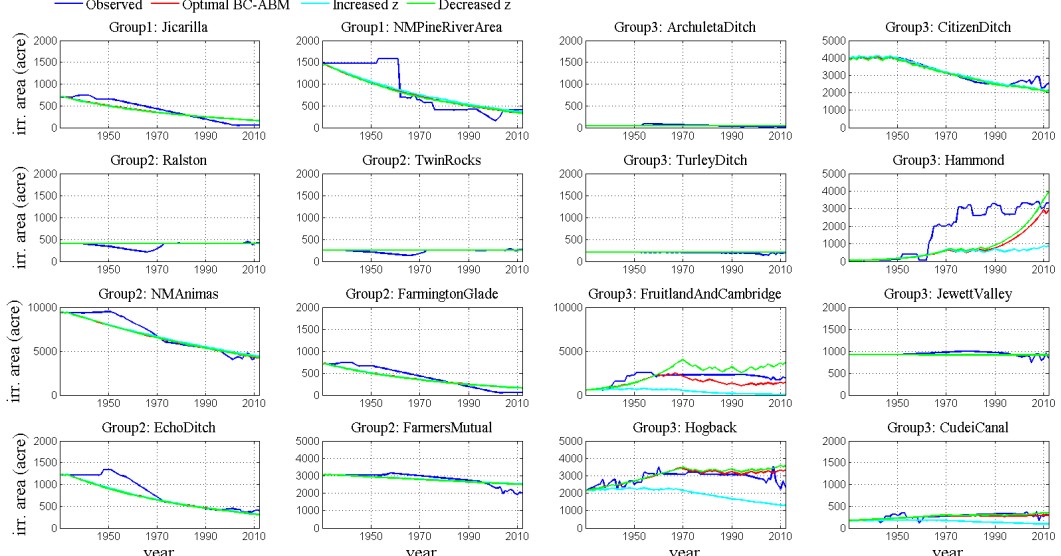

Figure 9. Irrigation area changes of each agents under the scenario of increasing (cyan) and
decreasing (green) $z$. The calibrated results (baseline simulation) are shown in red and
observations are shown in blue.