# Peer review of "Manuscript under review for journal Hydrol. Earth Syst. Sci."

_Hydrology and Earth System Sciences, 2018_

## Referee Comment (RC1) · Anonymous Referee #1 · 3 Feb 2019

This study aims to demonstrate that a hybrid modeling approach, coupling agent-based, Bayesian, cost-loss, and reservoir management models, yields a more representative simulation of stream flow and changes in irrigated area of the San Juan River Basin. To achieve this, the authors model the interaction between farmer agents and a river routing and reservoir management model. Individual behaviors of farmers are developed using the Theory of Planned Behavior, accomplished by applying Bayesian and cost-loss modeling approaches. In modeling coupled natural-human systems, developing hybrid models to utilize the advantages of each is an interesting approach. However, in its current state, I believe the manuscript needs substantial revision before I can recommend it for publication.

[Figure]

My greatest objection to the manuscript is that it needs to be better focused. The Introduction wanders in both scope and topic. Additionally, the research objectives are not clearly defined. The study area has clearly defined conflicts for water supply; however, these are not brought up until the Discussion section. Defining the importance of this study earlier in the manuscript is important to justify the research. The Results section contains a significant amount of interpretation that needs to be moved to the Discussion section. Finally, the Conclusion section doesn't address the greater impacts of the study.

Another concern is that although the authors have a substantial amount of time-series data, they make no attempt to reserve some of their data in order to validate their model. I would like to see an attempt or justification as to why no validation was conducted.

More detail on specific sections below:

Introduction: The Introduction section is too long and needs to be condensed. In its current form, the Introduction does not funnel from general to specific information relevant to the study and instead gets bogged down by the history of each modeling approach. Perhaps reframe in the following way: water policy challenges in a CNHS (including uncertainty) -> ABM -> TPB, the quickly explain how BI and CL will address the three components of TPB.

The last paragraph of the Introduction should clearly lay out all of the objectives of the paper. For example: It is the purpose of the study to demonstrate the utility of TPB in modeling human decision-making by 1) evaluating the impacts of uncertain risk perception on agent behavior, 2) comparing model results with conventional agent behavior rules, etc. . ..

Structure the Results section in the same order for ease of comprehension.

Methodology: The description of the Bayesian Inference Mapping section suffers from

excessive detail in regards to the manipulation of the Bayesian equations. Some of this should be moved into the Supplemental Materials.

In the agents' decision-making methodology, the authors are calculating the farmer's beliefs for each direct link in the cognitive map, but only choosing one link (resulting from the most extreme variable) to insert into the cost-loss model. Since the authors are not examining the joint probability of making a decision from all preceding factors, it is disingenuous to describe the methodology with the Bayesian network presented. Instead the authors should be explicit in their methodology that they are examining each preceding factor independently. This can be accomplished by describing the model's decision-making process at the beginning of Section 2.2 with the aid of Figure 1. The authors should note in their discussion that by only limiting decision-making to one preceding factor, the agents cannot respond to the cumulative effects of their environmental conditions.

In the calculation of "extremity" of environmental factors, the authors use the distance of the current value from half of the max value. If the mean of a variable is greater than that, most the of extremities will be artificial inflated. Using outliers within a variables' natural distribution of values will yield a more accurate characterization of extremities.

The authors need to be more explicit in describing the decision model of individual agents. How did the authors determine which factors were important in the agent's decision to increase/decrease irrigation area?

Case Study: This section (3.1) should be moved to the beginning of the methodology. Precipitation, NIIP Diversion, and Flow Violation are the main factors in your decision network; however, you do not describe their characteristics (mean, standard deviation) in your study area.

Section 3.2 and 3.3 should be relabeled to define it as the setup conditions of the coupled model.

Results: The methodology of the comparative study is introduced in the third paragraph of this section. It should be moved to the methodology section, described sufficiently, and stated as an objective of the paper, or removed entirely.

This section should be strictly limited to presenting the results of the model; however, the authors spend a significant amount of time interpreting the meaning of the results. These interpretations should be moved to the Discussion section.

Discussion: The authors introduce significant new information in the discussion section, particularly in regards to San Juan Basin water policy, that would be better served in the case study section. The conflict introduced here will help bring a sense of urgency to the research if presented earlier.

Conclusions: Since the authors used TPB to frame the human decision-making model, the authors should revisit TPB in regards to the successfulness of the approach.

Figures 5: The authors should explore whether presenting the data as a scatterplot will increase comprehension of model performance.

Specific Comments: The title is phrased awkwardly and not does give readers enough information on the content of the manuscript.

Table 1: Group 3b should be WITH shortage sharing
* * *

---

## Referee Comment (RC2) · Andres Baeza (Referee) · 6 Feb 2019

General Comments

In this work the authors developed an agent-based model to simulate agents that make decisions related to irrigation management. The agents consider climate and social information to update risk perception and cost of operations, to decide whether to increase or reduce water consumption for irrigation. The agents are located in a river network with a man-made structure that controls water flow. The results show that by considering this environmental and social information along with the perception of risk, agents can replicate water consumption patterns observed in the San Juan river

basin. I think this is a very interesting work that provides great methodological tools to develop a coupled hydrological, agent-based model. The introduction is clear and well supported by the literature. While some points could be made even clearer, the authors did a good job introducing the objectives and the methods proposed. I considered the method section to be the most interesting part of this paper. The Bayesian inference (BI) rule provides a great tool that combines robust math and easy applicability to develop the agents' decision-making framework. My main concern with the BI is the assumption or presumption of risk. In the model, when agents ignore incoming information, these agents are labeled as "risk-averse". I do not understand why, by not considering previous information, these agents would be considered risk-averse. My understanding is that risk-averse individuals pay more attention to not have great losses vs. a risk-seeking agent, who would give more importance or weight to potential large gains, thereby discounting loses. I think the authors need to clarify this point. Finally, for the methods, a sub-section containing the estimation and calibration methods, and the comparison with real data, is needed. Some aspects of these methods are described when the results were described later in the manuscript, and this created some confusion about the methods that were used.

The case study is quite interesting and well supported by time-series data. My main comment in this section is about the kind of agents their model is trying to simulate. It is not clear to me who are the "irrigated" and/or "ditch object" -agents. Are these infrastructure operators, managers, or a group of farmers with influence on the decision made to obtain water? The authors can do better explaining these agents. Another point that I think needs an explanation is how the social and climate factors that each agent considers as important were elicited. Some agents consider extreme precipitation, while others consider "animas precipitation". I also suggest that the authors differentiate between climate vs. social factors in Table 1. This would make the different socio-ecological factors that influence each agents' decisions clearer to the readers. I suggest looking at the ODD+D protocol, instead of ODD, to describe the model, because the ODD+D includes the decision-making aspect of the model (Müller et al.,

2013). The authors cited this study, but they have not used it. I consider the discussion to be somewhat weak and not in line with the aim of the study, nor the results. The discussion starts with a reflection about the policies implemented in the study area, but it was only loosely connected to the decisions of the agents, the information these agents considered, and the risk. There is no discussion or reflection about implementing theory-planned behavior, which I think would be a great step to incorporate real theories of human behavior into agent-based models. The authors should highlight this effort. Perhaps the discussion can be constructed around the following question: How do the risk perception, information flow, and costs influence policy outcomes in not only the San Juan river basin, but also in other basins? The discussion should start with a broader statement about the generality of the method and its applicability to other rivers. Then, it should include the implications of the results for policy outcomes, first for the example of the San Juan river, and then for other irrigated areas. Finally, the authors stated in 5.2 that they will discuss future research, yet no specific ideas were provided. In any case, these future directions should be included in the conclusion, rather than the discussion. At a minimum, a real discussion about these ideas, including what would be needed and other considerations, should be included.

Specific comments

Abstract I do not consider risk perception and uncertainty to be the same, as the author clearly described in the introduction (Line 107). On line 22, the authors should be more careful when introducing these terms in the abstract. Introduction Line 59: Why do the authors start with the word "therefore" to introduce planned behavior? Line 73: Need to introduce the low-cost rule. Line 89-100. In the abstract, the authors suggest that risk perception is included in the BI rule. They then introduce risk perception when discussing the CL rule. This causes some trouble understanding the model. Line 128: A line or two is needed stating what "two-way" coupling means. I think they refer to feedback between decisions, perception, and water dynamics. Is this correct?

Methods Line 229: A definition of subscripts i and j is needed.

[Figure]

Case Study Line 313: What does "cfs" stand for? What is this unit? Line 385: What does "matching" the time series mean? Is it based on Least Squares as a Maximum Likelihood? In other words, an explanation is needed on how the comparison between real data and simulated data was carried out. Line 418: An explanation for the Nash-Sutcliffe Efficiency is needed. Line 457: The phase including "...multi-objective calibration..." is not a result. This should be in the methods. Line 585: The statement beginning "The BC-ABM results ..." is also not a result. The fact that agents react to climate and socio-economic factors is part of the rules imposed by the model, but it is not a result per se. Line 624: I do not understand why the authors introduce multi-criteria decision analysis vs. other decision-making tools. It is an important tool, but it is hard to see the connection.

Figure 1: In ABM process 3, what is the question that leads to yes or no? It is related to the opportunity cost, but it needs to be stated in the figure. Figure 2: Perhaps a better name for "irrigated agents" is needed.

I hope these comments are useful to the authors.

---

## Short Comment (SC1) · 6 Mar 2019

**Reviewer 1**

This study aims to demonstrate that a hybrid modeling approach, coupling agent-based, Bayesian, cost-loss, and reservoir management models, yields a more representative simulation of stream flow and changes in irrigated area of the San Juan River Basin. To achieve this, the authors model the interaction between farmer agents and a river routing and reservoir management model. Individual behaviors of farmers are developed using the Theory of Planned Behavior, accomplished by applying Bayesian and cost-loss modeling approaches. In modeling coupled natural-human systems, developing hybrid models to utilize the advantages of each is an interesting approach. However, in its current state, I believe the manuscript needs substantial revision before I can recommend it for publication.

**Response**

We acknowledge the concern and suggestions from the reviewer. An itemized response of all the comments and the corresponding revisions is described as follow. Line numbers in this document correspond to the clean version (no track changes) of the revised draft.

My greatest objection to the manuscript is that it needs to be better focused. The Introduction wanders in both scope and topic. Additionally, the research objectives are not clearly defined. The study area has clearly defined conflicts for water supply; however, these are not brought up until the Discussion section. Defining the importance of this study earlier in the manuscript is important to justify the research. The Results section contains a significant amount of interpretation that needs to be moved to the Discussion section. Finally, the Conclusion section doesn't address the greater impacts of the study.

**Response**

We want to thank the reviewer for these specific suggestions. The introduction has been revised to focus on the challenges of simulating a coupled natural-human system (CNHS), the proposed solutions for water resources management using agent-based model (ABM) and finally using the theory of planned behavior (TPB) for ABM construction. We remove the details of uncertainty sources in the introduction to avoid the disturbance as suggested by Reviewer 2. We only keep some key equations in the Methodology section and move some supporting equations to the Supplemental Materials. The description of water conflict within the San Juan River Basin has been moved from the Discussion to the Case Study section to defining the water supply conflict earlier in the manuscript as suggested by both reviewers. We also add some potential application of the proposed method in the Conclusion.

Another concern is that although the authors have a substantial amount of time-series data, they make no attempt to reserve some of their data in order to validate their model. I would like to see an attempt or justification as to why no validation was conducted.

**Response**

We thank the comment from the reviewer. There are several reasons why we only show the calibration results. First, the final calibration range of parameter are relative narrow since we are aiming for the trend not the fluctuation of the irrigated area, the length of the record (as long as it is "long" enough) will not significantly affect the final outcomes. However, to response reviewer's suggestion, we put some example results from example

agent in this document. We use the first 65 years (1929-1993) to compute the first NSE and the rest 20 years to compute the second NSE. Since our purpose is to compare the BC-ABM and Non-BC-ABM, we did the same calculation for Non-BC-ABM as well. One can see that in both calibration and validation period, BC-ABM still outperform Non-BC-ABM.

[Figure]

Second, the fundamental assumption of calibration and validation is stationarity which might not be hold here especially when human decision is involved. This is also true for any traditional calibration and validation for process-based hydrologic model. If one known a significant land use and land cover change had occurred, the validation results will not hold. In our case, the unexpected external driver will significantly affect human behavior and violate the assumption of stationarity. This is also the main reason in the above four examples, the NSE value are lower in the validation period. In the San Juan Basin, the construction of the Navajo Reservoir is the obvious example of external driver change. However, the proposed BC-ABM can potentially allow agents to make adaptive decision by changing their lamda as well as by considering their neighbor's decision which is the suggested method to address the non-stationarity issue.

Third, the current Figure 3 is already a very "busy" figure which means we already have multiple lines, patterns and colors for various purpose and we need to shows that for all 16 agents. Add validation results (which means adding a number of new colors and/or patterns) will make the figure unreadable and might overwhelm our readers. This is the reason why only we show some example here for the reviewer.

More detail on specific sections below:
Introduction: The Introduction section is too long and needs to be condensed. In its current form, the Introduction does not funnel from general to specific information relevant to the study and instead gets bogged down by the history of each modeling approach. Perhaps reframe in the

following way: water policy challenges in a CNHS (including uncertainty) -> ABM -> TPB, the quickly explain how BI and CL will address the three components of TPB.

**Response**

We exactly follow this suggestion and revise the Introduction section. The description of challenges in CNHS model is in Line 33 to 41. We then move to the use of ABM for CNHS modeling in Line 42to 51. Line 52 to 66 describe how we proposed to use TPB to improve ABM that address uncertain risk perception and finally move to some description of using BI and CL to address three components of TPB in Line 67 to 96.

The last paragraph of the Introduction should clearly lay out all of the objectives of the paper. For example: It is the purpose of the study to demonstrate the utility of TPB in modeling human decision-making by 1) evaluating the impacts of uncertain risk perception on agent behavior, 2) comparing model results with conventional agent behavior rules, etc...

**Response**

As suggested by the reviewer, the following paragraph has been added to the end of the Introduction (Line 97 to 110) to clearly lay out all the objectives of this study:

"*To address these research gaps aforementioned, we developed an ABM based on the BI mapping and CL model, as an implementation of the TPB, and hereafter referred to as the "BC-ABM." The BC-ABM is "two-way" coupled with a river-routing and reservoir management model (RiverWare). Four objectives of this study are: 1) use the BC-ABM to quantify human decision considering uncertain risk perception, 2) demonstrate the improvement of BC-ABM compare to conventional agent behavior rules, 3) use the coupled BC-ABM-RiverWare to explicitly model the feedback loop between human and natural system and 4) test the BC-ABM-Riverware for different scenarios. The San Juan River Basin in New Mexico, USA is used as the demonstration basin for this effort. The calibrated BC-ABM-RiverWare model is used to evaluate the impacts of changing risk perception from all agents to the water management in this basin. In this study, multiple comparative experiments of conventional rule-based ABM (i.e., without using the BL and CL) are conducted to demonstrate the advantages of the proposed BC-ABM framework in modeling human decision-making processes. We also evaluate the effect of changing external economic conditions on an agent's decisions.*"

We also add a brief description of the study area: the San Juan River Basin in New Mexico in this last paragraph to give our readers an idea where we want to test the proposed method.

Structure the Results section in the same order for ease of comprehension.

**Response**

Given that we define this manuscript as a methodologic paper, the first three research objectives mentioned above are considered as the improvements of the ABM methodology. Therefore, we present results which are the demonstration of the ABM improvement in the last section of Case Study and follow the suggestion from the reviewer in the order of

objective 1 (quantify impacts of uncertain risk perception from agents), objective 2 (compare BC-ABM with conventional ABM) and objective 3 (demonstrate results from both BC-ABM and RiverWare to two-way coupling). Figure 3, 4 and 5 are used to visualize these results (Line 346 to 415).

The research objective 4 is considering as the application or pilot test of the BC-ABM-RiverWare. We present results from several different scenarios including extreme behaviors from agents as well as extreme socioeconomic driver change. Figure 6, 7, 8, and 9 are used to visualize these results (Line 416 to 504).

Methodology: The description of the Bayesian Inference Mapping section suffers from excessive detail in regards to the manipulation of the Bayesian equations. Some of this should be moved into the Supplemental Materials.

**Response**
The Methodology section has been revised by moving detailed derivations to the Supplemental Materials. We only keep the following equations from the original manuscript. Equation (3), (10), (11), (13), (15) and (16) are in the revised draft for the most critical parts of the BI mapping. We also keep the original Equation (17) for the determination of extremity and Equation (19) and (20) for the Cost-Lost model (Line 151 to 249). We add a section in the Supplemental Materials for the detailed method (Text S1).

In the agents' decision-making methodology, the authors are calculating the farmer's beliefs for each direct link in the cognitive map, but only choosing one link (resulting from the most extreme variable) to insert into the cost-loss model. Since the authors are not examining the joint probability of making a decision from all preceding factors, it is disingenuous to describe the methodology with the Bayesian network presented. Instead the authors should be explicit in their methodology that they are examining each preceding factor independently. This can be accomplished by describing the model's decision-making process at the beginning of Section 2.2 with the aid of Figure 1.

**Response**
The following sentence has been added to explicitly mention the associated assumption of using extremity from single instead of multiple factors in the BI mapping:

"*In this study, the extremity of each preceding factor is examined independently assuming each preceding factor is independent to each other (consider one not the joint probability of multiple factors in the BI mapping).*"

Please check Line 210 to 213. Also, following the reviewers' comment, we modify Figure 1 for this topic as well. Please check the new figure below and in the figure file.

[Figure]

The authors should note in their discussion that by only limiting decision-making to one preceding factor, the agents cannot respond to the cumulative effects of their environmental conditions. In the calculation of "extremity" of environmental factors, the authors use the distance of the current value from half of the max value. If the mean of a variable is greater than
that, most the of extremities will be artificial inflated. Using outliers within a variables' natural distribution of values will yield a more accurate characterization of extremities. The authors need to be more explicit in describing the decision model of individual agents. How did the authors determine which factors were important in the agent's decision to increase/decrease irrigation area?

**Response**

A more detailed explanation with an example has been added to Section 2.2.1 to explicitly describe the use of the extremity in the BI mapping in this study as suggested. Please check Line 213 to 216. The Limitation section is also expanded to include the exclusion of the joint probability of decision-making processes from all preceding factors caused the use of extremity in the BI mapping.

Case Study: This section (3.1) should be moved to the beginning of the methodology. Precipitation, NIIP Diversion, and Flow Violation are the main factors in your decision network; however, you do not describe their characteristics (mean, standard deviation) in your study area.

**Response**

Since the preceding factors described in Section 3.1 are specific for the case study area, they are not always true for other basins. Therefore, we keep the original paper structure for Section 3.1. However, we follow the reviewer's suggestion in three aspects. First, we add more example preceding factors in the methodology to give our reader a more concrete idea (Line 165 to 168). Second, in the new Table 1, we add the characteristics (mean, standard deviation) of these preceding factors as suggested by the reviewer. Please check

Table 1 below and the new table file. Third, we move part of the original Section 5.1 into this new Section 3.1 as suggested by the reviewer in the following comment. This provides a more informative background to our readers about the water conflict situation in the basin. Please check Line 275 to 295.

| Group | Number of agents | Factors considered in decision-making processes |
|---|---|---|
| **1.** (upstream of the Navajo Reservoir) | 2 | • mainstem upstream precipitation[c] (180.1 mm, 125.3 mm),
• the water level in the Navajo Reservoir[c] (6053.58 ft, 13.37 ft),
• number of flow violation at the outlet[c] (38.5, 38.8),
• cost-loss ratio[s] |
| **2.a** (Animas River without shortage sharing) | 5 | • tributary (Animas) precipitation[c] (79.2 mm, 38.2 mm),
• mainstem upstream precipitation[c] (180.1 mm, 125.3 mm),
• the water level in the Navajo Reservoir[c] (6053.58 ft, 13.37 ft),
• number of flow violation at the outlet[c] (38.5, 38.8),
• cost-loss ratio[s] |
| **2.b** (Animas River with shortage sharing) | 1 | • tributary (Animas) precipitation[c] (79.2 mm, 38.2 mm),
• mainstem upstream precipitation[c] (180.1 mm, 125.3 mm),
• the water level in the Navajo Reservoir[c] (6053.58 ft, 13.37 ft),
• number of flow violation at the outlet[c] (38.5, 38.8),
• shortage sharing[s],
• cost-loss ratio[s] |
| **3.a** (downstream of the Navajo Reservoir without shortage sharing) | 3 | • mainstem downstream precipitation[c] (82.9 mm, 96 mm),
• mainstem upstream precipitation[c] (180.1 mm, 125.3 mm),
• the water level in the Navajo Reservoir[c] (6053.58 ft, 13.37 ft),
• number of flow violation at the outlet[c] (38.5, 38.8),
• NIIP diversion[s] (159,310 ac-ft, 15131 ac-ft mm),
• cost-loss ratio[s] |
| **3.b** (downstream of the Navajo Reservoir with shortage sharing) | 5 | • mainstem downstream precipitation[c] (82.9 mm, 96 mm),
• mainstem upstream precipitation[c] (180.1 mm, 125.3 mm),
• the water level in the Navajo Reservoir[c] (6053.58 ft, 13.37 ft),
• number of flow violation at the outlet[c] (38.5, 38.8),
• NIIP diversion[s] (159,310 ac-ft, 15131 ac-ft mm),
• shortage sharing[s],
• cost-loss ratio[s] |

Section 3.2 and 3.3 should be relabeled to define it as the setup conditions of the coupled model.

**Response**
Follow the suggestion from the reviewer, we modify the title of Section 3.2 as "The BC-ABM-RiverWare Model Setup." We also move the model diagnose outcomes to the new Section 3.3 and modify the new title as "The BC-ABM-RiverWare Model Diagnostics." We present the new Section 3.3 following the order of our research objectives in the last paragraph of Introduction as recommended by the reviewer.

Results: The methodology of the comparative study is introduced in the third paragraph of this section. It should be moved to the methodology section, described sufficiently, and stated as an objective of the paper, or removed entirely.

**Response**

Following the paper reconstruction suggestion from both reviewers, we move this section into the Case Study part given that we use the historical data from the study area to make the comparative study. However, we do partly follow the reviewer's suggestion and provide a clearer description of the comparative study which is actually not a methodology. The conventional rule-based type, deterministic ABM is the mainstream of the agent-based model and we cite our previous work for this model (Line 385 to 388). The purpose of this comparison is to demonstrate that by introduction BI mapping and CL model, we can better capture the historical pattern and trend of the decision on irrigated area changes.

Also, follow the reviewer's suggestion, we explicitly stated this effort as one of the research objectives. Please check Line 101 to 102.

This section should be strictly limited to presenting the results of the model; however, the authors spend a significant amount of time interpreting the meaning of the results. These interpretations should be moved to the Discussion section.

**Response**

After the paper restructure process, the current Result section only shows two tested scenarios: the effect of changing agents' risk perception and the effect of changing socioeconomic condition. These scenarios have stronger policy implementation meanings rather than mathematical outcomes. Therefore, explanations are critical for these results to provide a meaning content rather than just describing the figures. We believe most of our readers, who are hydrologists or water resources scientists, not mathematicians, will be more interested in the hydrologic reasoning and can potentially inform water management policy. Please check Line 416 to 504.

Discussion: The authors introduce significant new information in the discussion section, particularly in regards to San Juan Basin water policy, that would be better served in the case study section. The conflict introduced here will help bring a sense of urgency to the research if presented earlier.

**Response**

We follow the suggestion and move a large part of the original Section 5.1, especially for the water conflict part to the Case Study section (Line 275 to 295). We keep the part that related to our modeling results in the revised Section 5.1 as a deeper discussion on the institutional context and other water policy related issues.

Conclusions: Since the authors used TPB to frame the human decision-making model, the authors should revisit TPB in regards to the successfulness of the approach.

**Response**

Following the suggestion from both reviewers, we revisit how our proposed BC-ABM can implement TPB in both the Discussion and Conclusion. Please check Line 506 to 522 and also Line 564 to 568.

Figures 5: The authors should explore whether presenting the data as a scatterplot will increase comprehension of model performance.

**Response**
Since both streamflow and reservoir release are time series, we do feel the line format is a better representation rather than the scatterplot. However, we agree with the reviewer that the original Figure 5 is a bit hard to read. We modify the pattern and the thickness of lines to improve the readability. We keep the color as blue: observation and red: modeling which matches the Figure 4 (calibration results). Please check the new figure below and the new figure file.

[Figure]

Specific Comments: The title is phrased awkwardly and not does give readers enough information on the content of the manuscript.

**Response**
The title has been changed to "*Using a coupled agent-based modeling approach to quantify risk perception in water management decisions*" to better reflect the modified content.

Table 1: Group 3b should be WITH shortage sharing

**Response**
The typo has been corrected.

---

## Short Comment (SC2) · 6 Mar 2019

We upload our response as a pdf file, please check the attachment. Thank you.

Please also note the supplement to this comment:
https://www.hydrol-earth-syst-sci-discuss.net/hess-2018-555/hess-2018-555-SC2-supplement.pdf
* * *

---

## Author Response (AR1)

**Editor**

I am satisfied with the manner in which the authors have responded to the comments of the two reviewers. The responses demonstrate that the authors have carefully and seriously considered the comments and suggestions, and provided plausible responses.

I agree with the comment of reviewer 1 that the title of the paper may need to be revised, and I am happy with the reply by the authors.

I therefore invite the authors to submit an improved version of the manuscript, consistent with the responses made to the two reviews. In addition, I would like the authors to consider also the following three minor comments from my side:

**Response**
We want to thank the Editor for the positive review comments and further improve the quality of this draft.

1. I have one question not raised by the two reviewers, and it relates to what the manuscript states in lines 363-366:
"Note that the information of winter precipitation … was gathered from NOAA ground-based rainfall monitoring gauges where we used the coming year's winter precipitation as a proxy for the snowpack forecast in the causal structure."
This might seem to suggest perfect knowledge on future water conditions, which might seem to be inconsistent with the purpose of the paper, namely to include uncertainty in water management decisions. I invite the authors to explain how this information was included in their modelling framework.

**Response**
The winter precipitation data from NOAA is perfect knowledge in the current modeling structure. However, the among of snowpack that will be resulted is uncertain. We modify the manuscript and clarify this part. Please check Line 325 to 330. This setting also offer an opportunity if the actual precipitation forecast data become available, we can easily replace the NOAA precipitation with the forecast precipitation.

2. I also invite the authors to address several quite odd typos and grammatical inaccuracies in the manuscript (e.g. in lines 38, 56, 1012, 102, 120, 127, 132, 192, 315, 380, 539, 558, 561).

**Response**
We correct all these English error in the revised manuscripts. Thank you for the careful review.

3. I also want to point out that SI units are used in HESS. So please also give the SI equivalents (m2 or ha, m, m3 and m3/s) of acre, feet, acre-feet and cfs when used in the text. I did not understand the unit "ac-ft mm" used in the revised Table 1. No clue what it means. Kindly explain. Note that the "manuscript preparation guidelines for authors" explicitly states that "The use of SI units or SI-derived units is mandatory." (https://www.hydrology-and-earth-system-sciences.net/for_authors/manuscript_preparation.html)

**Response**

SI units have been add to all English units in the manuscript. The conversion we used are: 1 arce-ft = 1234 m$^3$, 1 acre = 4046 m$^2$, 1 cfs = 0.0283 cms, 1 ft =0.3048 m,

Success!

Pieter van der Zaag

**Reviewer 1**

This study aims to demonstrate that a hybrid modeling approach, coupling agent-based, Bayesian, cost-loss, and reservoir management models, yields a more representative simulation of stream flow and changes in irrigated area of the San Juan River Basin. To achieve this, the authors model the interaction between farmer agents and a river routing and reservoir management model. Individual behaviors of farmers are developed using the Theory of Planned Behavior, accomplished by applying Bayesian and cost-loss modeling approaches. In modeling coupled natural-human systems, developing hybrid models to utilize the advantages of each is an interesting approach. However, in its current state, I believe the manuscript needs substantial revision before I can recommend it for publication.

**Response**

We acknowledge the concern and suggestions from the reviewer. An itemized response of all the comments and the corresponding revisions is described as follow. Line numbers in this document correspond to the clean version (no track changes) of the revised draft.

My greatest objection to the manuscript is that it needs to be better focused. The Introduction wanders in both scope and topic. Additionally, the research objectives are not clearly defined. The study area has clearly defined conflicts for water supply; however, these are not brought up until the Discussion section. Defining the importance of this study earlier in the manuscript is important to justify the research. The Results section contains a significant amount of interpretation that needs to be moved to the Discussion section. Finally, the Conclusion section doesn't address the greater impacts of the study.

**Response**

We want to thank the reviewer for these specific suggestions. The introduction has been revised to focus on the challenges of simulating a coupled natural-human system (CNHS), the proposed solutions for water resources management using agent-based model (ABM) and finally using the theory of planned behavior (TPB) for ABM construction. We remove the details of uncertainty sources in the introduction to avoid the disturbance as suggested by Reviewer 2. We only keep some key equations in the Methodology section and move some supporting equations to the Supplemental Materials. The description of water conflict within the San Juan River Basin has been moved from the Discussion to the Case Study section to defining the water supply conflict earlier in the manuscript as suggested by both reviewers. We also add some potential application of the proposed method in the Conclusion.

Another concern is that although the authors have a substantial amount of time-series data, they make no attempt to reserve some of their data in order to validate their model. I would like to see an attempt or justification as to why no validation was conducted.

**Response**

We thank the comment from the reviewer. There are several reasons why we only show the calibration results. First, the final calibration range of parameters are relative narrow since we are aiming for the trend not the annual fluctuation of the irrigated area, the length of the record (as long as it is "long" enough) will not significantly affect the final outcomes. However, to response reviewer's suggestion, we put some results from example agent in this document. We use the first 65 years (1929-1993) to compute the first NSE and the rest 20 years to compute the second NSE. Since our purpose is to compare the BC-ABM and Non-BC-ABM, we did the same calculation for

Non-BC-ABM as well. One can see that in both pre-1993 and the post-1993 period, BC-ABM still outperform Non-BC-ABM.

[Figure]

Second, the fundamental assumption of calibration and validation is stationarity which might not be hold here especially when human decision is involved. This is also true for any traditional calibration and validation procedure in process-based hydrologic models. If one know a significant land use and land cover change had occurred, the validation results will not (and should not) hold. In our case, the unexpected external driver will significantly affect human behavior and violate the assumption of stationarity. This is also the main reason in the above four examples, the NSE value are lower in the post-1993 period. In the San Juan Basin, the construction of the Navajo Reservoir is the obvious example of external driver change. However, the proposed BC-ABM can potentially allow agents to make adaptive decision by changing their lamda as well as by considering their neighbor's decision which is the suggested method to address the non-stationarity issue.

Third, the current Figure 3 is already a very "busy" figure which means we already have multiple lines, patterns and colors for various purpose and we need to shows that for all 16 agents. Add validation results (which means adding a number of new colors and/or patterns) will make the figure unreadable and might overwhelm our readers. This is the reason why only we show some example here for the reviewer.

More detail on specific sections below:

Introduction: The Introduction section is too long and needs to be condensed. In its current form, the Introduction does not funnel from general to specific information relevant to the study and instead gets bogged down by the history of each modeling approach. Perhaps reframe in the following way: water policy challenges in a CNHS (including uncertainty) -> ABM -> TPB, the quickly explain how BI and CL will address the three components of TPB.

**Response**

We exactly follow this suggestion and revise the Introduction section. The description of challenges in CNHS model is in Line 33 to 41. We then move to the use of ABM for CNHS modeling in Line 42to 51. Line 52 to 66 describe how we proposed to use TPB to improve ABM that address uncertain risk perception and finally move to some description of using BI and CL to address three components of TPB in Line 67 to 96.

The last paragraph of the Introduction should clearly lay out all of the objectives of the paper. For example: It is the purpose of the study to demonstrate the utility of TPB in modeling human decision-making by 1) evaluating the impacts of uncertain risk perception on agent behavior, 2) comparing model results with conventional agent behavior rules, etc…

**Response**

As suggested by the reviewer, the following paragraph has been added to the end of the Introduction (Line 97 to 110) to clearly lay out all the objectives of this study:

*"To address these research gaps aforementioned, we developed an ABM based on the BI mapping and CL model, as an implementation of the TPB, and hereafter referred to as the "BC-ABM." The BC-ABM is "two-way" coupled with a river-routing and reservoir management model (RiverWare). Four objectives of this study are: 1) use the BC-ABM to quantify human decision considering uncertain risk perception, 2) demonstrate the improvement of BC-ABM compare to conventional agent behavior rules, 3) use the coupled BC-ABM-RiverWare to explicitly model the feedback loop between human and natural system and 4) test the BC-ABM-Riverware for different scenarios. The San Juan River Basin in New Mexico, USA is used as the demonstration basin for this effort. The calibrated BC-ABM-RiverWare model is used to evaluate the impacts of changing risk perception from all agents to the water management in this basin. In this study, multiple comparative experiments of conventional rule-based ABM (i.e., without using the BL and CL) are conducted to demonstrate the advantages of the proposed BC-ABM framework in modeling human decision-making processes. We also evaluate the effect of changing external economic conditions on an agent's decisions."*

We also add a brief description of the study area: the San Juan River Basin in New Mexico in this last paragraph to give our readers an idea where we want to test the proposed method.

Structure the Results section in the same order for ease of comprehension.

**Response**

Given that we define this manuscript as a methodologic paper, the first three research objectives mentioned above are considered as the improvements of the ABM methodology. Therefore, we present results which are the demonstration of the ABM improvement in the last section of Case Study and follow the suggestion from the reviewer in the order of objective 1 (quantify impacts of uncertain risk perception from agents), objective 2 (compare BC-ABM with conventional ABM) and objective 3 (demonstrate results from both BC-ABM and RiverWare to two-way coupling). Figure 3, 4 and 5 are used to visualize these results (Line 350 to 419).

The research objective 4 is considering as the application or pilot test of the BC-ABM-RiverWare. We present results from several different scenarios including extreme behaviors from agents as well as extreme socioeconomic driver change. Figure 6, 7, 8, and 9 are used to visualize these results (Line 420 to 508).

Methodology: The description of the Bayesian Inference Mapping section suffers from excessive detail in regards to the manipulation of the Bayesian equations. Some of this should be moved into the Supplemental Materials.

**Response**

The Methodology section has been revised by moving detailed derivations to the Supplemental Materials. We only keep the following equations from the original manuscript. Equation (3), (10), (11), (13), (15) and (16) are in the revised draft for the most critical parts of the BI mapping. We also keep the original Equation (17) for the determination of extremity and Equation (19) and (20) for the Cost-Lost model (Line 151 to 249). We add a section in the Supplemental Materials for the detailed method (Text S1).

In the agents' decision-making methodology, the authors are calculating the farmer's beliefs for each direct link in the cognitive map, but only choosing one link (resulting from the most extreme variable) to insert into the cost-loss model. Since the authors are not examining the joint probability of making a decision from all preceding factors, it is disingenuous to describe the methodology with the Bayesian network presented. Instead the authors should be explicit in their methodology that they are examining each preceding factor independently. This can be accomplished by describing the model's decision-making process at the beginning of Section 2.2 with the aid of Figure 1.

**Response**

The following sentence has been added to explicitly mention the associated assumption of using extremity from single instead of multiple factors in the BI mapping:

"*In this study, the extremity of each preceding factor is examined independently assuming each preceding factor is independent to each other (consider one not the joint probability of multiple factors in the BI mapping).*"

Please check Line 210 to 213. Also, following the reviewers' comment, we modify Figure 1 for this topic as well. Please check the new figure below and in the figure file.

[Figure]

The authors should note in their discussion that by only limiting decision-making to one preceding factor, the agents cannot respond to the cumulative effects of their environmental conditions. In the calculation of "extremity" of environmental factors, the authors use the distance of the current value from half of the max value. If the mean of a variable is greater than
that, most the of extremities will be artificial inflated. Using outliers within a variables' natural distribution of values will yield a more accurate characterization of extremities. The authors need to be more explicit in describing the decision model of individual agents. How did the authors determine which factors were important in the agent's decision to increase/decrease irrigation area?

**Response**
A more detailed explanation with an example has been added to Section 2.2.1 to explicitly describe the use of the extremity in the BI mapping in this study as suggested. Please check Line 213 to 216. The Limitation section is also expanded to include the exclusion of the joint probability of decision-making processes from all preceding factors caused the use of extremity in the BI mapping.

Case Study: This section (3.1) should be moved to the beginning of the methodology. Precipitation, NIIP Diversion, and Flow Violation are the main factors in your decision network; however, you do not describe their characteristics (mean, standard deviation) in your study area.

**Response**
Since the preceding factors described in Section 3.1 are specific for the case study area, they are not always true for other basins. Therefore, we keep the original paper structure for Section 3.1. However, we follow the reviewer's suggestion in three aspects. First, we add more example preceding factors in the methodology to give our reader a more concrete idea (Line 165 to 168). Second, in the new Table 1, we add the characteristics (mean, standard deviation) of these preceding factors as suggested by the reviewer. Please check Table 1 below and the new table file. Third, we move part of the original Section 5.1 into this new Section 3.1 as suggested by the reviewer in the following comment. This provides a more informative background to our readers about the water conflict situation in the basin. Please check Line 275 to 295.

| Group | Number of agents | Factors considered in decision-making processes |
|---|---|---|
| **1.** (upstream of the Navajo Reservoir) | 2 | • mainstem upstream precipitation[c] (180.1 mm, 125.3 mm),
• the water level in the Navajo Reservoir[c] (1845 m, 4.07 m),
• number of flow violation at the outlet[c] (38.5, 38.8),
• cost-loss ratio[s] |
| **2.a** (Animas River without shortage sharing) | 5 | • tributary (Animas) precipitation[c] (79.2 mm, 38.2 mm),
• mainstem upstream precipitation[c] (180.1 mm, 125.3 mm),
• the water level in the Navajo Reservoir[c] (1845 m, 4.07 m),
• number of flow violation at the outlet[c] (38.5, 38.8),
• cost-loss ratio[s] |
| **2.b** (Animas River with shortage sharing) | 1 | • tributary (Animas) precipitation[c] (79.2 mm, 38.2 mm),
• mainstem upstream precipitation[c] (180.1 mm, 125.3 mm),
• the water level in the Navajo Reservoir[c] (1845 m, 4.07 m),
• number of flow violation at the outlet[c] (38.5, 38.8),
• shortage sharing[s],
• cost-loss ratio[s] |
| **3.a** (downstream of the Navajo Reservoir without shortage sharing) | 3 | • mainstem downstream precipitation[c] (82.9 mm, 96 mm),
• mainstem upstream precipitation[c] (180.1 mm, 125.3 mm),
• the water level in the Navajo Reservoir[c] (1845 m, 4.07 m),
• number of flow violation at the outlet[c] (38.5, 38.8),
• NIIP annual diversion[s] (0.197 billion $m^3$, 0.019 billion $m^3$),
• cost-loss ratio[s] |
| **3.b** (downstream of the Navajo Reservoir with shortage sharing) | 5 | • mainstem downstream precipitation[c] (82.9 mm, 96 mm),
• mainstem upstream precipitation[c] (180.1 mm, 125.3 mm),
• the water level in the Navajo Reservoir[c] (1845 m, 4.07 m),
• number of flow violation at the outlet[c] (38.5, 38.8),
• NIIP annual diversion[s] (0.197 billion $m^3$, 0.019 billion $m^3$),
• shortage sharing[s],
• cost-loss ratio[s] |

Section 3.2 and 3.3 should be relabeled to define it as the setup conditions of the coupled model.

**Response**
Follow the suggestion from the reviewer, we modify the title of Section 3.2 as "The BC-ABM-RiverWare Model Setup." We also move the model diagnose outcomes to the new Section 3.3 and modify the new title as "The BC-**ABM-RiverWare Model** Diagnostics." We present the new Section 3.3 following the order of our research objectives in the last paragraph of Introduction as recommended by the reviewer.

Results: The methodology of the comparative study is introduced in the third paragraph of this section. It should be moved to the methodology section, described sufficiently, and stated as an objective of the paper, or removed entirely.

**Response**

Following the paper reconstruction suggestion from both reviewers, we move this section into the Case Study part given that we use the historical data from the study area to make the comparative study. However, we do partly follow the reviewer's suggestion and provide a clearer description of the comparative study which is actually not a methodology. The conventional rule-based type, deterministic ABM is the mainstream of the agent-based model and we cite our previous work for this model (Line 385 to 388). The purpose of this comparison is to demonstrate that by introduction BI mapping and CL model, we can better capture the historical pattern and trend of the decision on irrigated area changes.

Also, follow the reviewer's suggestion, we explicitly stated this effort as one of the research objectives. Please check Line 101 to 102.

This section should be strictly limited to presenting the results of the model; however, the authors spend a significant amount of time interpreting the meaning of the results. These interpretations should be moved to the Discussion section.

**Response**
After the paper restructure process, the current Result section only shows two tested scenarios: the effect of changing agents' risk perception and the effect of changing socioeconomic condition. These scenarios have stronger policy implementation meanings rather than mathematical outcomes. Therefore, explanations are critical for these results to provide a meaning content rather than just describing the figures. We believe most of our readers, who are hydrologists or water resources scientists, not mathematicians, will be more interested in the hydrologic reasoning and can potentially inform water management policy. Please check Line 420 to 508.

Discussion: The authors introduce significant new information in the discussion section, particularly in regards to San Juan Basin water policy, that would be better served in the case study section. The conflict introduced here will help bring a sense of urgency to the research if presented earlier.

**Response**
We follow the suggestion and move a large part of the original Section 5.1, especially for the water conflict part to the Case Study section (Line 275 to 295). We keep the part that related to our modeling results in the revised Section 5.1 as a deeper discussion on the institutional context and other water policy related issues.

Conclusions: Since the authors used TPB to frame the human decision-making model, the authors should revisit TPB in regards to the successfulness of the approach.

**Response**
Following the suggestion from both reviewers, we revisit how our proposed BC-ABM can implement TPB in both the Discussion and Conclusion. Please check Line 510 to 526 and also Line 568 to 572.

Figures 5: The authors should explore whether presenting the data as a scatterplot will increase comprehension of model performance.

**Response**

Since both streamflow and reservoir release are time series, we do feel the line format is a better representation rather than the scatterplot. However, we agree with the reviewer that the original Figure 5 is a bit hard to read. We modify the pattern and the thickness of lines to improve the readability. We keep the color as blue: observation and red: modeling which matches the Figure 4 (calibration results). Please check the new figure below and the new figure file.

[Figure]

Specific Comments: The title is phrased awkwardly and not does give readers enough information on the content of the manuscript.

**Response**
The title has been changed to "*Using a coupled agent-based modeling approach to quantify risk perception in water management decisions*" to better reflect the modified content.

Table 1: Group 3b should be WITH shortage sharing

**Response**
The typo has been corrected.

**Reviewer 2**

General Comments

In this work the authors developed an agent-based model to simulate agents that make decisions related to irrigation management. The agents consider climate and social information to update risk perception and cost of operations, to decide whether to increase or reduce water consumption for irrigation. The agents are located in a river network with a man-made structure that controls water flow. The results show that by considering this environmental and social information along with the perception of risk, agents can replicate water consumption patterns observed in the San Juan river basin. I think this is a very interesting work that provides great methodological tools to develop a coupled hydrological, agent-based model.

The introduction is clear and well supported by the literature. While some points could be made even clearer, the authors did a good job introducing the objectives and the methods proposed.

**Response**

We want to thank the reviewer for these constructive comments and suggestions which greatly improve the clarity of the entire manuscript. We further condense the Introduction section following the comment from Reviewer 1. Line numbers in this document correspond to the clean version (no track changes) of the revised draft.

I considered the method section to be the most interesting part of this paper. The Bayesian inference (BI) rule provides a great tool that combines robust math and easy applicability to develop the agents' decision-making framework. My main concern with the BI is the assumption or presumption of risk. In the model, when agents ignore incoming information, these agents are labeled as "risk-averse". I do not understand why, by not considering previous information, these agents would be considered risk-averse. My understanding is that risk-averse individuals pay more attention to not have great losses vs. a risk-seeking agent, who would give more importance or weight to potential large gains, thereby discounting loses. I think the authors need to clarify this point.

**Response**

We agree with the reviewer for the definition of risk-averse. In our reasoning, we define "risk-averse" as "*do not trust the new incoming information because it could be uncertain and rather to stick with her/his own experience*" In other words, an agent is not taking any risk by changing its behavior. The sentence has been modified in the revised draft, please check Line 194 to 196.

Finally, for the methods, a sub-section containing the estimation and calibration methods, and the comparison with real data, is needed. Some aspects of these methods are described when the results were described later in the manuscript, and this created some confusion about the methods that were used.

**Response**

One sentence has been added to the last paragraph of the Methodology section to remind our readers the model diagnose and the comparison with real data in the case study area (Line 248 to 249).

Both reviewers suggest moving the text for the model diagnose part to the earlier section of the manuscript. Given that the model calibration process requires historical data from the case study area as references, we need to put this section after we describe the case study area. Therefore, we decide to put the original Section 4.1 as the last section of the Case Study in the revised draft. The reasons are two-fold. First, since it is out of the Result section, the confusion that Reviewer 2 described can be avoided. Second, this paper structure rearrangement will follow the suggestion from Reviewer 1 that the order of the outcome presentation is the same as the order of research objectives stated in the last paragraph of Introduction. Please check Line 350 to 419.

The case study is quite interesting and well supported by time-series data. My main comment in this section is about the kind of agents their model is trying to simulate. It is not clear to me who are the "irrigated" and/or "ditch object" -agents. Are these infrastructure operators, managers, or a group of farmers with influence on the decision made to obtain water? The authors can do better explaining these agents.

**Response**
Agents are groups of farmers in our study. In the RiverWare model set up, they are quantified as several "water use objects" which we named them as irrigation ditches. These agents (or irrigation ditch) are an aggregation of farmers in that specific area and our assumption is that since USBR aggregate these farmers into several single entities, they will make similar management decision in reality. We add an additional explanation in Section 3.2, please check Line 308 to 311.

Another point that I think needs an explanation is how the social and climate factors that each agent considers as important were elicited. Some agents consider extreme precipitation, while others consider "animas precipitation". I also suggest that the authors differentiate between climate vs. social factors in Table 1. This would make the different socio-ecological factors that influence each agents' decisions clearer to the readers.

**Response**
Precipitation is a preceding factor candidate of all agents. However, depending on the geographical location of agents, they need to consider precipitation at different locations. For example, upstream agents (e.g. the Group 1 in our case) do not need to consider downstream precipitation since that will not affect their water availability. This is an advantage of using ABM which the spatial heterogeneity can be addressed in the model. We add a sentence in the new Section 3.2 to better explain this and also follow the reviewer's suggestion add a description about how climatic and social factor might affect agents' decision. Please check Line 316 to 340.

Also following the reviewer's suggestion, we modify Table 1 with a superscript that distinguishes climatic and social factors. Please check the new Table 1 below and in the new table file.

| Group | Number of agents | Factors considered in decision-making processes |
|---|---|---|
| **1.** (upstream of the Navajo Reservoir) | 2 | • mainstem upstream precipitation[c] (180.1 mm, 125.3 mm),
• the water level in the Navajo Reservoir[c] (1845 m, 4.07 m),
• number of flow violation at the outlet[c] (38.5, 38.8),
• cost-loss ratio[s] |
| **2.a** (Animas River without shortage sharing) | 5 | • tributary (Animas) precipitation[c] (79.2 mm, 38.2 mm),
• mainstem upstream precipitation[c] (180.1 mm, 125.3 mm),
• the water level in the Navajo Reservoir[c] (1845 m, 4.07 m),
• number of flow violation at the outlet[c] (38.5, 38.8), |

| | | |
|---|---|---|
| | | • cost-loss ratio[s] |
| **2.b** (Animas River with shortage sharing) | 1 | • tributary (Animas) precipitation[c] (79.2 mm, 38.2 mm),
• mainstem upstream precipitation[c] (180.1 mm, 125.3 mm),
• the water level in the Navajo Reservoir[c] (1845 m, 4.07 m),
• number of flow violation at the outlet[c] (38.5, 38.8),
• shortage sharing[s],
• cost-loss ratio[s] |
| **3.a** (downstream of the Navajo Reservoir without shortage sharing) | 3 | • mainstem downstream precipitation[c] (82.9 mm, 96 mm),
• mainstem upstream precipitation[c] (180.1 mm, 125.3 mm),
• the water level in the Navajo Reservoir[c] (1845 m, 4.07 m),
• number of flow violation at the outlet[c] (38.5, 38.8),
• NIIP annual diversion[s] (0.197 billion $m^3$, 0.019 billion $m^3$),
• cost-loss ratio[s] |
| **3.b** (downstream of the Navajo Reservoir with shortage sharing) | 5 | • mainstem downstream precipitation[c] (82.9 mm, 96 mm),
• mainstem upstream precipitation[c] (180.1 mm, 125.3 mm),
• the water level in the Navajo Reservoir[c] (1845 m, 4.07 m),
• number of flow violation at the outlet[c] (38.5, 38.8),
• NIIP annual diversion[s] (0.197 billion $m^3$, 0.019 billion $m^3$),
• shortage sharing[s],
• cost-loss ratio[s] |

I suggest looking at the ODD+D protocol, instead of ODD, to describe the model, because the ODD+D includes the decision-making aspect of the model (Müller et al., 2013). The authors cited this study, but they have not used it.

**Response**

We modify our ODD document into the ODD+D format, please check the new supplement materials

 I consider the discussion to be somewhat weak and not in line with the aim of the study, nor the results. The discussion starts with a reflection about the policies implemented in the study area, but it was only loosely connected to the decisions of the agents, the information these agents considered, and the risk. There is no discussion or reflection about implementing theory-planned behavior, which I think would be a great step to incorporate real theories of human behavior into agent-based models. The authors should highlight this effort. Perhaps the discussion can be constructed around the following question: How do the risk perception, information flow, and costs influence policy outcomes in not only the San Juan river basin, but also in other basins? The discussion should start with a broader statement about the generality of the method and its applicability to other rivers. Then, it should include the implications of the results for policy outcomes, first for the example of the San Juan river, and then for other irrigated areas.

**Response**

Following the suggestion form both reviewers, we completely restructure the Discussion section. First, most of the original Section 5.1 has been moved to the Case Study which provides a more informative background to our readers about the water conflict situation in the basin. Please check Line 275 to 295. Second, we change the title of the section to "Generalized the modeling framework and policy implementation for other basins" and start this section with a broader statement about the generality of the method and how it addresses the challenges of how the proposed BA-ABM implementing the TPB as a first step to incorporate real theories of human behavior into agent-based models. Please check Line 510 to 526. Third, we use our results in the San Juan River as an example to explain the models' applicability for policy implementation. Please check Line 527 to 539.

Finally, the authors stated in 5.2 that they will discuss future research, yet no specific ideas were provided. In any case, these future directions should be included in the conclusion, rather than the discussion. At a minimum, a real discussion about these ideas, including what would be needed and other considerations, should be included.

**Response**
We change the title of the revised Section 5.2 as "Model limitations" which we only use this section to discuss the limitations of the current draft such as data availability and model structure in BI mapping as well as extremity. Please check Line 540 to 561.

Following the suggestion from both reviewers, we move the ideas of future research into the Collusion section (Line 586 to 594).

Specific comments
Abstract
I do not consider risk perception and uncertainty to be the same, as the author clearly described in the introduction (Line 107). On line 22, the authors should be more careful when introducing these terms in the abstract.

**Response**
From revised the abstract following reviewer's suggestion and more specifically talk about risk perception (Line 10 to 28).

Introduction
Line 59: Why do the authors start with the word "therefore" to introduce planned behavior?

**Response**
The word "therefore" is deleted.

Line 73: Need to introduce the low-cost rule.

**Response**
We improve the description of the CL model (including the calculation of taking action based on low-cost concept) in Methodology section given that the Introduction is only intended to provide a high-level idea of what is CL model. Please check Line 217 to 239 in the Methodology section.

Line 89-100. In the abstract, the authors suggest that risk perception is included in the BI rule. They then introduce risk perception when discussing the CL rule. This causes some trouble understanding the model.

**Response**
Line 90 to 96 was previous studies and directly use the CL model for the risk perception. However, as we summarized in Line 94 to 96, this previous study did not provide a detailed methodology for parameter determination and ignore spatial heterogeneity. Therefore, we want to improve this aspect by the proposed method of using BI mapping to quantify risk perception given that the BI mapping can explicitly consider conditional probability. We highlight this as a gap in the last paragraph of the Introduction section (Line 100 to 104).

Line 128: A line or two is needed stating what "two-way" coupling means. I think they refer to feedback between decisions, perception, and water dynamics. Is this correct?

**Response**
Since we reorganize the Introduction section, we only briefly mention this term "two-way coupling" in the Introduction section. We provide a detailed description of what we mean by "two-way coupling" in Section 2.1. Please check Line 137 to 143.

Methods
Line 229: A definition of subscripts i and j is needed.

**Response**
We move this equation to the Supplements Materials as suggested by Reviewer 1. We add the following sentence in the Text S1 to explain "$i$" and "$j$:"

"$i$ is the index for the preceding factor and $j$ is the index for the management behavior"

Case Study
Line 313: What does "cfs" stand for? What is this unit?

**Response**
Add "cubic feet per second." Please check Line 266.

Line 385: What does "matching" the time series mean? Is it based on Least Squares as a Maximum Likelihood? In other words, an explanation is needed on how the comparison between real data and simulated data was carried out.

**Response**
We reword this as "recreate" the historical trend. Please check Line 348 to 353.

Line 418: An explanation for the Nash-Sutcliffe Efficiency is needed.

**Response**
The Nash-Sutcliffe Efficiency is widely used in water resource to assess the predictive power of process-based models. We add the original citation into the manuscript (Line 381).

Nash, J. E.; Sutcliffe, J. V. (1970). "River flow forecasting through conceptual models part I — A discussion of principles". Journal of Hydrology. 10 (3): 282–290.

Line 457: The phase including "…multi-objective calibration:…" is not a result. This should be in the methods.

**Response**

As we mentioned above, we did a complete paper structure reconstruction following the suggestions from both reviewers. This sentence does not fit in the revised draft and has been removed.

Line 585: The statement beginning "The BC-ABM results…" is also not a result. The fact that agents react to climate and socio-economic factors is part of the rules imposed by the model, but it is not a result per se.

**Response**

Since we restructure the paper, the entire sentence has been removed.

Line 624: I do not understand why the authors introduce multicriteria decision analysis vs. other decision-making tools. It is an important tool, but it is hard to see the connection.

**Response**

The text about multi-criteria decision analysis in both the Discussion and Conclusion section has been removed completely.

Figure 1: In ABM process 3, what is the question that leads to yes or no? It is related to the opportunity cost, but it needs to be stated in the figure.

**Response**

We update Figure 1 following suggestions from both reviewers. Please check the new Figure 1 below and in the new figure file.

[Figure]

Figure 2: Perhaps a better name for "irrigated agents" is needed.

**Response**

We update Figure 2 following the suggestion from the reviewer. Please check the new Figure 2 below and in the new figure file.

[Figure]

I hope these comments are useful to the authors.

**Response**
Again, we want to thank the reviewer for these constructive comments and suggestions which greatly improve the clarity of the entire manuscript.

[revised manuscript text omitted]

**Commented [EY14]:** Reviewer 1, Detailed comment: extremity as a single factor.

**Commented [EY15]:** Reviewer 1, Detailed comment: extremity example.

as socioeconomic conditions. The CL model is applied in this study to address this concern. The

CL model measures the tendency of an adverse event affecting the decision of whether to take costly precautionary action to protect oneself against losses from that event.  Based on the theory of Cost-Benefit Analysis, the probability of taking an action $p$ is related to the expected cost of taking action $C$ and opportunity lost of not taking the action $L$:

$$p \geq \frac{C}{L} = z \qquad (9)$$

where $z$ is defined as the Cost-Loss (CL) ratio and only when this value is less the probability of the event occurring, the precautionary action will be taken.

To fit the CL model into the proposed ABM framework, we modify the above CL model following the concept of Tena and Gómez (2008) and Matte et al. (2017) which added the perception of risk into the decision-making process. We define "$C$" as the expected cost of taking management action that will potentially increase the gross economic profit and "$L$" as the expected opportunity loss of not taking such management action. The CL ratio ($z$), as a measure of tendency, can be compared with the prior belief of an agent's for taking a management decision ($\Gamma_{pr}^{t}$ in

Equation 7). When $\Gamma_{pr}^{t}$ is greater than $z$, this decision will become real world management action since it makes economic senses.

$$\Gamma_{pr}^{t} \geq z = \frac{C}{L} = \frac{the\ expected\ cost\ of\ taking\ management\ action}{opportunity\ loss\ of\ not\ taking\ management\ action} \qquad (10)$$

When $z$ increases, it means the cost of taking management action goes up or the opportunity loss of not taking management action goes down. In either case, agents are less likely to take action due to reduced profits. When $z$ decreases, following the same logic, agents are more likely to take action.

**Commented [EY16]:** Reviewer 1, Detailed comment: move detailed methodology into supplemental materials.

**Commented [EY17]:** Reviewer 2, Specific comment: CL model

Figure 1 summarizes the methodology in Section 2.2 applied to this study. Agent's decision-making and action process will start when receiving information ($\Gamma_{pd}^t$) from RiverWare and the conditional probability of its decision $\Gamma_{pr}^t$ will be computed after the most "highly recognized" preceding factor is decided by the $V_i$ values. This probability of an agent's decision will be compared with the CL ratio ($z$) to account for the external economic conditions where the agent is located. The final management action from the agent will depend on whether the probability of making a decision for an agent's is greater (take the action) or smaller (do not take the action) than the CL ratio. This process is repeated annually throughout the entire simulation period. We will use the case study to demonstrate the capability of this proposed method and diagnose the model with the historical data.

> **Commented [EY18]:** Reviewer 2, Major comment: Methodology.

**3. Case Study**

**3.1. Background of the Study Area**

[revised manuscript text omitted]

---

## Author Response (AR2)

Editor:
Dear authors,

Thanks for the revised version of the manuscript. Although the manuscript now looks coherent, I still have the following remaining observations, which I kindly request you to address. This involves minor work and can be quickly done.

1. I could not find the Table and the Figures. Kindly submit these, as I need to check these.

**Response**
Following the instruction from the journal staff, we submit two pdf files: 1. A clean version of the manuscript with tables and figures. And 2. A mark-up version of the manuscript with this response document.

2. Title: you have indeed changed the title, but now I ask myself: is the objective to quantify risk perception or to analyse the role of uncertainty and risk perception in water management decisions? If the latter this needs to be reflected in the title. Could the following resolve this:

 "Using a coupled agent-based modeling approach to analyse the role of risk perception in water management decisions"
**Response**
We want to thank the editor for a further improved title.

3. There must be a brief paragraph explaining the structure of the paper towards the end of section 1 (Introduction, after line 110), as you had done in the original version (lines 143-147)
**Response**
We add a paragraph at the end of the introduction section to describe the structure of the paper.

4. The following editorial issues need to be addressed (with line numbers indicated):
L35: that subject to -> that is subject to
L67: Implementating -> Implementing
L78: uncertainties
L98: implementation
L133: coupled -> couple
L148: setting -> setup
L154: delete "taking"
L194: do -> does
L195: to stick -> sticks
L226: less the probability -> less than the probability
L268: water consumption -> water consumer
L325-326: "… was not taken from RiverWare; rather was gathered from NOAAA ..." isn't the following formulation much more straightforward: "… was taken from NOAAA ..."
L329: "winter perception": do you mean "winter precipitation"?
L332: "proceeding factors that considered by all …" do you mean "preceding factors considered by all …"?
L369: manual -> manually
L372: with larger irrigated area first -> with the largest irrigated areas first
L421: "creditability": do you mean "credibility"?

L447: the total violation days still decrease -> the total number of violation days still decreases

L497: "stockholders": is this the correct word? Or do you mean "stakeholders"?

L522-523: "the data required for the model diagnose such as long-term historical area time series might not..": do you mean: ""the data required for model diagnostics and calibration, such as long-term historical area time series, might not.."?

L551: by a detailed interview with decision makers -> by detailed interviews with decision makers

L555: from single preceding factor -> from a single preceding factor

L581: confirm -> confirms

L587: its -> their

L591: … reservoir as different type of agents -> .. reservoir as agents

L592: agent -> agents

**Response**

We want to thank the editor for these editorial suggestions. We corrected them all and marked them in the mark-up version.

[revised manuscript text omitted]